# $Re^2$: Unlocking LLM Reasoning via Reinforcement Learning with Re-solving

**Pinzheng Wang**[1]  **Shuli Xu**[1]  **Juntao Li**[1]*  **Yu Luo**[2]  **Dong Li**[2]  **Jianye Hao**[3]  **Min Zhang**[1]

[1]School of Computer Science and Technology, Soochow University
[2]Department of Foundation Model, 2012 Labs, Huawei
[3]College of Intelligence and Computing, Tianjin University

```
{pzwang1, slxu}@stu.suda.edu.cn
{ljt, minzhang}@suda.edu.cn
{luoyu81, lidong106}@huawei.com
jianye.hao@tju.edu.cn
```

## Abstract

Reinforcement learning with verifiable rewards (RLVR) has shown promise in enhancing the reasoning performance of large language models (LLMs) by increasing test-time compute. However, even after extensive RLVR training, such models still tend to generate unnecessary and low-quality steps in their chain-of-thought (CoT), leading to inefficient overthinking and lower answer quality. We show that when the initial direction or quality of the CoT is suboptimal, the model often fails to reach the correct answer, even after generating several times more tokens than when the initial CoT is well-initialized. To this end, we introduce *Reinforcement Learning with Re-solving* (**Re**$^2$), in which LLMs learn to flexibly abandon unproductive reasoning paths and restart the solution process when necessary, rather than always committing to a final answer. Re$^2$ applies pure reinforcement learning without any preliminary supervised fine-tuning, successfully amplifying the rare redo behavior in vanilla models from only 0.5% to over 30%. This leads to substantial performance gains over standard RLVR under the same training compute budget, and also demonstrates notable improvements in test-time performance as the number of samples increases.

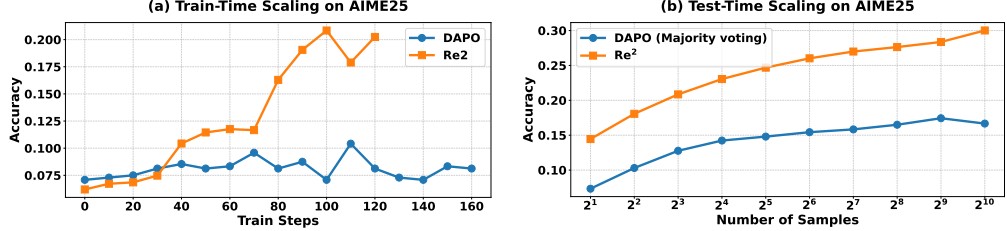

Figure 1: **(a)** Accuracy improvements of **DAPO** and **Re**$^2$ on Qwen2.5-7B-Instruct at each training step, with comparable numbers of generated and trained tokens per step. **(b)** Test-time scaling of **DAPO** and **Re**$^2$ under the same training budget, as the number of samples increases.

## 1 Introduction

Recent studies have shown that large language models (LLMs) can achieve strong reasoning abilities through scaling test-time compute (Snell et al., 2024; Wu et al., 2025). By generating longer chains of thought (CoTs) that incorporate planning, reflection, and self-correction, LLMs attain higher accuracy on complex reasoning tasks such as coding and mathematics (Yang et al., 2025a; Bercovich et al.,

---
*Corresponding author

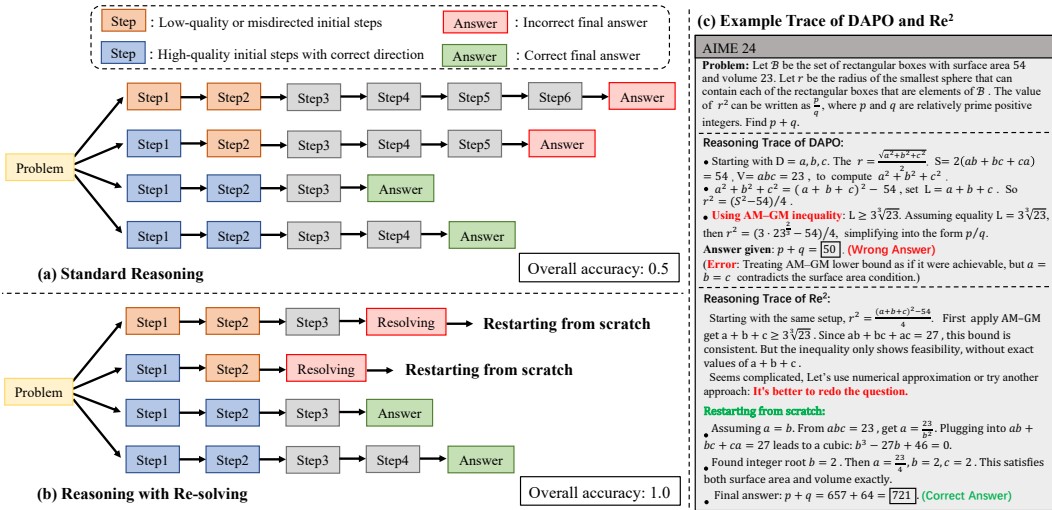

Figure 2: Illustration of reasoning with re-solving. (a) In standard reasoning, when the initial steps are suboptimal, the LLM struggles to reach the correct answer even after generating more reasoning steps and tokens. (b) Reasoning with re-solving, allowing the model to abandon an unpromising path and restart at any point, leads to a higher overall accuracy. (c) Example trace of DAPO and $Re^2$: Both models initially attempt an incorrect approach using the AM–GM inequality; however, $Re^2$ detects the failure, restarts, and arrives at the correct answer.

2025; Team et al., 2025; Wu et al., 2024). To this end, state-of-the-art models adopt reinforcement learning (RL) in post-training, which has proven effective in producing longer CoTs and strengthening deep-thinking capabilities (Guo et al., 2025; Shao et al., 2024; Yu et al., 2025).

However, even with extensive RL training, LLMs still suffer from issues such as overthinking (Chen et al., 2024; Cuadron et al., 2025) and underthinking (Wang et al., 2025; Cuesta-Ramirez et al., 2025; Ding et al., 2025), generating unnecessary or low-quality reasoning steps that degrade both efficiency and overall performance. In this paper, we investigate the limitations of test-time scaling in existing LLMs by analyzing the correlation between CoT length and accuracy, together with the impact of early reasoning quality on final outcomes. Our analysis reveals that when the initial reasoning steps are suboptimal, LLMs struggle to reach the correct answer, even after generating significantly more reasoning steps and tokens (Figure 2(a)). However, solving complex problems often requires early exploration, during which the model may pursue directions that initially appear promising but ultimately lead to dead ends or errors. Motivated by these findings, we aim to enable models to restart the reasoning process when the current path proves unproductive, thereby improving overall accuracy by escaping unpromising paths (Figure 2(b)).

To equip models with this capability, we introduce *Reinforcement Learning with Re-solving* ($Re^2$), a novel framework that equips models with the ability to either finalize an answer or re-solve the problem. During training, the model extends partial reasoning trajectories and learns to decide whether to continue or to restart from its current progress. If the model chooses to re-solve, its trajectory receives a reward equal to the expected success rate of solving the problem from scratch. If it instead produces a final answer, the reward is assigned as $1$ for a correct solution and $0$ otherwise. Under this scheme, when the current reasoning prefix is in the wrong direction or of low quality, abandoning it and re-solving the problem is more likely to yield higher accuracy and thus a larger reward. Conversely, when the reasoning trajectory is promising, directly producing the final answer leads to a higher expected reward. Notably, $Re^2$ requires no preliminary supervised fine-tuning. Through pure reinforcement learning alone, it amplifies the rare redo behavior in vanilla models from just 0.5% to over 30%, thereby enabling models to flexibly decide when to re-solve and ultimately leading to more accurate reasoning, as illustrated in Figure 2(c).

We evaluate $Re^2$ on a diverse set of reasoning benchmarks, including AIME 2024, AIME 2025 (MAA Committees), AMC 2023 (AI-MO, 2024), GSM8K (Cobbe et al., 2021), and GPQA-

Diamond (Rein et al., 2021), covering a wide range of domains and difficulty levels. Our evaluation covers five models ranging from 3B to 14B parameters, including pre-trained, instruction-tuned, and reasoning models. Experimental results demonstrate that our approach achieves significant improvements in reasoning performance compared to recent RLVR methods such as DAPO (Yu et al., 2025) under the same training budget. Moreover, $Re^2$ yields a superior trade-off curve between test-time compute and performance compared to majority voting (Wang et al., 2022). We believe our work highlights the promise of integrating RL with a new paradigm of *re-solving* reasoning, which goes beyond the traditional single-chain approach and opens up new directions for developing more flexible and reliable reasoning in LLMs.

## 2 RELATED WORK

**LLM for reasoning.** Existing approaches enhance the reasoning capabilities of LLMs through prompt engineering (Wei et al., 2022; Yang et al., 2023), supervised fine-tuning (Yang et al., 2024a; Qin et al., 2024), and reinforcement learning (Schulman et al., 2017; Shao et al., 2024; Zheng et al., 2025; Yue et al., 2025; Zhang et al., 2025; Wang et al., 2024). Among these, reinforcement learning with verifiable rewards (RLVR) has emerged as a mainstream paradigm for post-training optimization, encouraging models to produce longer CoTs with planning and self-reflection, thereby pushing the frontier of reasoning performance (OpenAI, 2024; Qu et al., 2024; Gandhi et al., 2024; Zeng et al., 2025). Nevertheless, even after extensive RLVR training, LLMs remain prone to overthinking and underthinking, leading to redundant or low-quality reasoning steps (Chen et al., 2024; Cuadron et al., 2025; Wang et al., 2025; Cuesta-Ramirez et al., 2025). This remains a fundamental limitation of the prevailing paradigm, in which the model generates a single CoT trajectory and ultimately derives its final answer within that trajectory (Wen et al., 2025; Shojaee et al., 2025). Recent studies (Yang et al., 2025b; Fu et al., 2025) have attempted to address this issue by backtracking to earlier steps or terminating low-confidence reasoning chains, but these methods are limited to supervised fine-tuning or decoding strategies and do not leverage the potential of RL. To the best of our knowledge, our work is the first to propose a reasoning paradigm that allows models to abandon unproductive reasoning paths and re-solve problems from scratch through reinforcement learning.

**Test-time scaling.** Recent studies have shown that LLMs can effectively improve reasoning performance by increasing inference-time compute (Snell et al., 2024; Welleck et al., 2024; Wu et al., 2025; Muennighoff et al., 2025). Large reasoning models, as exemplified by OpenAI's O1, learn to produce traces that are longer than the typical solutions via SFT or RLVR (OpenAI, 2024; Qin et al., 2024; Guo et al., 2025; Zhao et al., 2024). Some works improve performance by allowing models to iteratively revise their answers through multiple rounds of self-correction (Xiong et al., 2025; Zhao et al., 2025; Xi et al., 2024; Paul et al., 2024; Yang et al., 2024b). In addition, parallel sampling methods (e.g., majority voting (Wang et al., 2022; Wan et al., 2024) and tree search (Hao et al., 2023; Zhang et al., 2024)) further improve performance by increasing the number of samples. In contrast, our approach scales test-time compute by enabling the model to abandon unpromising reasoning trajectories and re-solve the problem when necessary, thereby unlocking the potential of test-time scaling for reasoning.

## 3 DIFFICULTY OF RECOVERING FROM SUBOPTIMAL EARLY REASONING

To gain a deeper understanding of the limitations of test-time scaling in existing LLMs, we first analyze the correlation between CoT length and reasoning accuracy (Section 3.1), and further investigate why LLMs fail even when they having sufficient capabilities (Section 3.2). In our experiments, we select Qwen2.5-7B-Instruct (Yang et al., 2024a) as a representative instruction-tuned LLM and DeepScaleR-1.5B-Preview (Luo et al., 2025) as a representative long-CoT reasoning model extensively trained with RLVR. We evaluate them on AMC23 and AIME25, respectively, which aligns the difficulty of datasets with the capabilities of each model.

### 3.1 CORRELATION BETWEEN RESPONSE LENGTH AND PERFORMANCE

Although training models to produce longer CoTs can significantly improve reasoning performance, we aim to examine whether, for the same problem across multiple samples, longer CoTs actually lead

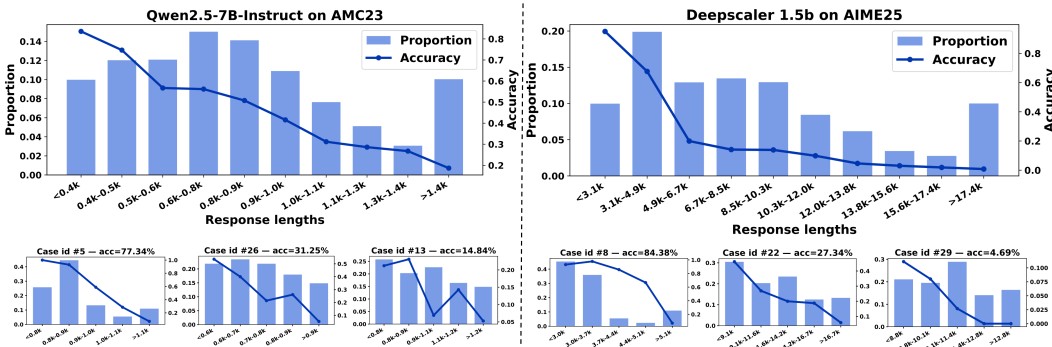

Figure 3: Correlation between CoT length and reasoning performance. The bars represent the proportion of cases within different CoT length intervals, while the line indicates the average accuracy for responses in each interval. The large plots in the top row show the correlation across the entire dataset. The small plots in the second row illustrate the correlation at the level of individual problems.

to higher accuracy. To this end, we analyze the correlation between response length and accuracy by sampling 128 responses per problem. As shown in Figure 3, our results reveal **a clear negative correlation between CoT length and accuracy**, both across the entire dataset and at the level of individual problems of varying difficulty. We further conduct case analysis of CoTs with different lengths for the same problem, which shows that longer responses are typically caused by early critical mistakes, such as following the wrong solution path or overanalyzing the problem's assumptions, making recovery unlikely regardless of the number of additional tokens generated. Additional results are provided in Appendix E.

## 3.2 IMPACT OF INITIAL REASONING QUALITY ON FINAL ACCURACY

As shown in Section 3.1, shorter CoTs are typically more likely to be correct for a given problem. We hypothesize that this is because suboptimal early reasoning leads the model to generate longer responses, yet recovery is still challenging. To illustrate this phenomenon, we design an experiment in which we truncate different proportions (20%, 40%, 60%, 80%) of **incorrect responses** and prompt LLMs to continue reasoning from these prefixes. We then measure the average accuracy of completions from each truncated prefix and compute the relative drop in accuracy compared to reasoning from scratch on the original problem. As shown in Figure 4, for each prefix length we report the number of cases where the relative drop in accuracy exceeds the 25% or 75% threshold ("All Drops"). The results show that as prefix length increases, the relative drop becomes larger, indicating that **the longer a model continues along an incorrect trajectory, the more likely it is to fail**.

Furthermore, we investigate how early such performance degradation begins in an incorrect response. For each response, we record the shortest prefix at which the relative drop first exceeds the threshold ("First Drops"). The results reveal that for most incorrect responses, a significant drop in accuracy already occurs when only the first 20% of the response is used as the prefix. This demonstrates that **once early reasoning is misguided, the model rarely recovers and struggles to return to the correct reasoning path**.

## 3.3 TAKEAWAYS

Based on the above analysis, we conclude that: **(1)** For a given problem, shorter responses following smoother reasoning tend to achieve higher accuracy, whereas longer responses are often associated with lower accuracy; **(2)** The quality of early reasoning process is crucial for the final accuracy.

## 4 RE$^2$: REINFORCEMENT LEARNING WITH RESOLVING

The above results highlight that a coherent reasoning process and high-quality early reasoning are crucial for model performance. However, when tackling challenging problems, it is often unavoidable

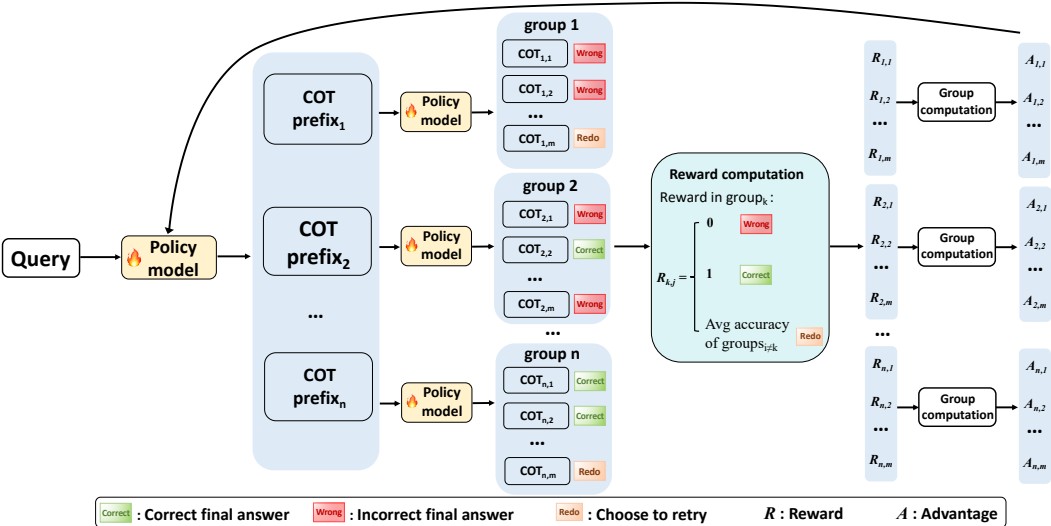

Figure 4: Number of cases where the relative drop exceeds the threshold when continuing from different proportions of incorrect response prefixes, compared to reasoning from scratch. "All Drops" counts all such cases at each prefix proportion, while "First Drops" records the earliest prefix for each response where the drop exceeds the threshold.

Figure 5: The framework of $Re^2$. For each query, $Re^2$ samples multiple prefixes, then generates multiple continuations for each prefix. The advantage is calculated within each group, while the out-of-group accuracy is used as the reward for the redo action.

for models to explore early reasoning paths that initially appear promising but ultimately fail to yield correct solutions. LLMs are also prone to generating low-quality content during the initial stages of reasoning. Similar to how humans may reconsider their strategy when solving difficult problems, we aim to equip models with the ability to restart the reasoning process when the current trajectory appears unpromising or leads to confusion. To this end, we propose $\mathbf{Re}^2$, which leverages reinforcement learning to train models to flexibly re-solve problems during reasoning.

### 4.1 PREFIX GROUP GENERATION

Recent RLVR methods such as GRPO (Shao et al., 2024) and DAPO (Yu et al., 2025) improve pass@1 accuracy by sampling multiple reasoning trajectories in parallel for each query and rewarding only those that yield correct answers. In contrast, $Re^2$ aims to improve the final answer quality by allowing the model to rationally abandon an ongoing reasoning path and restart from scratch when the trajectory appears confusing or unpromising. Meanwhile, our method requires estimating the success rate of re-solving from scratch, which provides the reward signal that guides the model to learn whether to redo or to commit to a final answer. To this end, we first sample $n$ full responses for each query. Each response is randomly truncated at a proportion uniformly drawn from $[0, 0.8]$, producing $n$ diverse prefixes that serve as intermediate reasoning states. For each prefix, the model generates $m$ CoT continuations, and all continuations derived from the same prefix are grouped together for subsequent advantage calculation, as illustrated in Figure 5. To incentivize the resolve capability of the base model, we design a specialized prompting strategy, described in Appendix A.

## 4.2 Reward Strategy with Re-solving

To encourage the model to rationally abandon unpromising reasoning paths and restart from scratch when necessary, we design a reward strategy that explicitly incorporates the option to re-solve. For the $j$-th CoT continuation of the $i$-th prefix $\text{Pre}_i$, denoted as $O_{i,j}$, there are three possible outcomes $C_{i,j}$: providing a correct final answer ($C_{i,j} = \texttt{correct}$), providing an incorrect final answer ($C_{i,j} = \texttt{incorrect}$), or choosing to re-solve the problem ($C_{i,j} = \texttt{resolve}$).

For continuations that yield final answers, the reward assignment follows standard RLVR: the model receives a reward of 1 for a correct answer and 0 otherwise. For continuations that choose to re-solve, the reward is given by the expected accuracy of re-solving from scratch, estimated using out-of-group CoT completions, i.e., completions whose prefix is not $\text{Pre}_i$. Specifically, let $P_{\neq i}(\texttt{correct})$, $P_{\neq i}(\texttt{incorrect})$, and $P_{\neq i}(\texttt{resolve})$ denote the empirical probabilities of correct, incorrect, and resolve outcomes among the $(n-1) \times m$ out-of-group continuations. When at most $R$ redo rounds are allowed, the reward is given by[1]:

$$r_{i,j} = \begin{cases} 1, & \text{if } C_{i,j} = \texttt{correct} \\ 0, & \text{if } C_{i,j} = \texttt{incorrect} \\ P_{\neq i}(\texttt{correct}) \cdot \frac{1 - P_{\neq i}(\texttt{resolve})^R}{1 - P_{\neq i}(\texttt{resolve})}, & \text{if } C_{i,j} = \texttt{resolve} \end{cases} \tag{1}$$

This three-way reward strategy encourages the model to continue reasoning when the current trajectory is promising, and to re-solve when the trajectory is confused or flawed, since the expected accuracy of re-solving exceeds that of continuation.

## 4.3 Advantage Computation and Parameter Update

---

**Algorithm 1** $\textbf{Re}^2$: Reinforcement Learning with Resolving

---

**Input** initial policy model $\pi_\theta$; task prompts $\mathcal{D}$; maximum training steps $s$; number of prefixes $n$; number of continuations $m$; max resolve rounds $R$; clipping thresholds $\varepsilon_{\texttt{low}}, \varepsilon_{\texttt{high}}$; update steps per batch $\mu$

**for** step = 1,...,$s$ **do**
1:   Sample a mini-batch $\mathcal{D}_b$ from $\mathcal{D}$
2:   For each question $q \in \mathcal{D}_b$, sample $n$ responses from $\pi_{\theta_{\text{old}}}(\cdot \mid q)$ and truncate them at random ratios to form prefixes $\{\text{Pre}_i\}_{i=1}^n$
3:   For each prefix $\text{Pre}_i$, sample $m$ continuations $\{O_{i,j}\}_{j=1}^m \sim \pi_{\theta_{\text{old}}}(\cdot \mid q, \text{Prefix}_i)$
4:   Compute rewards $\{r_{i,j}\}$ according to Eq. 1
5:   Filter out degenerate groups if all $\{r_{i,j}\}_{j=1}^m$ are identical
6:   Compute group-wise advantages $\hat{A}_{i,j}$ for each continuation according to Eq. 2
7:   **for** iteration = 1,...,$\mu$ **do**
         Update the policy model $\pi_\theta$ by maximizing $\mathcal{J}_{\text{Re}^2}(\theta)$ (Eq. 3)
8:   Update the old policy $\pi_{\theta_{\text{old}}} \leftarrow \pi_\theta$
**Output** updated policy $\pi_\theta$

---

After computing rewards under the re-solving strategy, we compute group-wise advantages and update the policy parameters following DAPO (Yu et al., 2025). Specifically, the advantage for the $j$-th continuation $O_{i,j}$ of prefix $\text{Pre}_i$ is defined as the reward normalized by subtracting the group mean and dividing by the group standard deviation:

$$\hat{A}_{i,j} = \frac{r_{i,j} - \text{mean}(\{r_{i,j}\}_{j=1}^m)}{\text{std}(\{r_{i,j}\}_{j=1}^m)}. \tag{2}$$

If all continuations within a group yield the same outcome (all correct, all incorrect, or all choosing resolve), then $\hat{A}_{i,j} = 0$ and the gradients vanish. Such degenerate groups are filtered out during

---

[1] The detailed derivation is provided in Appendix B.

training. The computed advantage is then broadcast to all response tokens of the corresponding continuation.

Formally, for each query $q \in \mathcal{D}$, we first sample $n$ prefixes $\{\text{Pre}_i\}_{i=1}^n \sim \pi_{\theta_{\text{old}}}(\cdot \mid q)$. For each prefix $\text{Pre}_i$, we then sample $m$ continuations $\{O_{i,j}\}_{j=1}^m \sim \pi_{\theta_{\text{old}}}(\cdot \mid q, \text{Pre}_i)$. The optimization objective is:

$$\mathcal{J}_{\text{Re}^2}(\theta) = \mathbb{E}_{[q \sim \mathcal{D}, \{\text{Pre}_i\}_{i=1}^n \sim \pi_{\theta_{\text{old}}}(\cdot|q), \{O_{i,j}\}_{j=1}^m \sim \pi_{\theta_{\text{old}}}(\cdot|q,\text{Pre}_i)]}$$

$$\left[ \frac{1}{nm} \sum_{i=1}^n \sum_{j=1}^m \frac{1}{|O_{i,j}|} \sum_{t=1}^{|O_{i,j}|} \min \left( \frac{\pi_\theta^{i,j,t}}{\pi_{\theta_{\text{old}}}^{i,j,t}} \hat{A}_{i,j}, \ \text{clip}\left( \frac{\pi_\theta^{i,j,t}}{\pi_{\theta_{\text{old}}}^{i,j,t}}, \ 1 - \varepsilon_{\text{low}}, \ 1 + \varepsilon_{\text{high}} \right) \hat{A}_{i,j} \right) \right], \quad (3)$$

where $\pi^{i,j,t} = \pi(O_{i,j,t} \mid q, \text{Pre}_i, O_{i,j,<t})$ denotes the conditional probability of the $t$-th token in continuation $O_{i,j}$ given the query $q$ and the prefix $\text{Pre}_i$. The models $\pi_\theta$ and $\pi_{\theta_{\text{old}}}$ correspond to the training policy and the sampling policy, respectively. The clipping thresholds $\varepsilon_{\text{low}}$ and $\varepsilon_{\text{high}}$ are hyperparameters used to bound the importance sampling ratio for stable optimization. The overall training algorithm is summarized in Algorithm 1.

## 5 EXPERIMENTS

### 5.1 EXPERIMENTAL SETUP

**Training datasets.** We construct our training set from the DAPO-Math-17K dataset (Yu et al., 2025), which is collected from AoPS[2] and official competition sources. The dataset covers a wide range of mathematical domains and contains 17K problems in total. To ensure reliable rule-based reward signals and minimize parsing errors, all answers are transformed into integers.

**Baselines & Models.** We compare $\text{Re}^2$ against the vanilla model (before RL training) and the recent RLVR method DAPO (Yu et al., 2025), which we follow for advantage computation and parameter updates. To ensure a fair comparison, both methods are trained with the same amount of generated tokens during RL optimization. To evaluate the effectiveness of $\text{Re}^2$ across model types and scales, we conduct experiments on both base and instruction-tuned LLMs, including Qwen-7B-Base, Qwen-14B-Base, Llama-3.2-3B-Instruct, and Qwen2.5-7B-Instruct (Dubey et al., 2024; Yang et al., 2024a). We further evaluate on DeepSeek-R1-Distill-Llama-8B (Guo et al., 2025), a reasoning model specifically finetuned to generate long chains of thought.

**Benchmarks.** To comprehensively evaluate the reasoning ability of our model, We adopt five widely used benchmarks covering diverse difficulty levels and domains: **AIME 2024** (MAA Committees) contains 30 challenging problems from the 2024 American Invitational Mathematics Examination. The exam is designed to test advanced problem-solving skills across algebra, geometry, combinatorics, number theory, and probability, and is often used as a challenging benchmark for evaluating reasoning ability in language models. **AIME 2025** (MAA Committees) follows the same format as AIME 2024, with 30 comparably difficult problems. As the most recent edition, it reduces the risk of contamination from pretraining or post-training data. **AMC 2023** (AI-MO, 2024) consists of 40 problems covering algebra, geometry, number theory, and combinatorics. Compared to the AIME benchmarks, its difficulty level is relatively lower. **GSM8K** (Cobbe et al., 2021) is a curated dataset of 1,319 elementary-level math word problems. Each problem typically requires two to eight reasoning steps, primarily involving multi-step arithmetic, making it a standard benchmark for assessing fundamental mathematical reasoning. **GPQA** (Rein et al., 2021) is a challenging dataset of graduate-level questions in physics, biology, and chemistry, where even PhD-level domain experts achieve only around 69.7% accuracy. In our experiments, we use the highest-quality subset, **GPQA-Diamond**, which consists of 198 carefully selected questions designed to provide a rigorous test of advanced scientific reasoning.

**Training and evaluation details.** For $\text{Re}^2$, we use a learning rate of $1 \times 10^{-6}$. Each training step processes a batch of 32 queries, with $n = 8$ prefixes sampled per query and $m = 8$ continuations generated for each prefix. The maximum sequence length is set to 8192 tokens. The clipping

---

[2]https://artofproblemsolving.com/

Table 1: Experimental results on five reasoning benchmarks. $Re^2$ consistently improves the overall reasoning performance of each model over DAPO ($p$-value $< 0.05$). Red numbers in parentheses indicate performance gains relative to DAPO.

| Models | Methods | AIME24 | AIME25 | AMC23 | GSM8K | GPQA | Avg |
|---|---|---|---|---|---|---|---|
| Base Model | | | | | | | |
| Qwen2.5-7B Base | + DAPO | 11.9 | 10.3 | 64.7 | 91.8 | 29.7 | 41.7 |
| | + $Re^2$ | **17.1** | **19.0** | **70.8** | **93.6** | **36.8** | **47.5** (+5.8) |
| Qwen2.5-14B Base | + DAPO | 18.2 | 15.7 | 64.0 | 94.3 | 44.8 | 47.4 |
| | + $Re^2$ | **28.5** | **23.4** | **68.5** | **94.6** | **49.6** | **52.9** (+5.5) |
| Instruct Model | | | | | | | |
| Llama3.2-3B-Instruct | None | 6.2 | 0.4 | 23.0 | 67.2 | 2.7 | 19.9 |
| | + DAPO | 15.0 | 0.5 | 32.3 | 80.4 | **20.7** | 29.8 |
| | + $Re^2$ | **17.7** | **2.8** | **38.4** | **83.2** | 20.2 | **32.5** (+2.7) |
| Qwen2.5-7B-Instruct | None | 11.4 | 7.5 | 51.4 | 85.3 | 33.4 | 37.8 |
| | + DAPO | 16.0 | 8.6 | 62.3 | 92.6 | 35.4 | 43.0 |
| | + $Re^2$ | **18.6** | **21.2** | **64.7** | **94.1** | **38.4** | **47.4** (+4.4) |
| Reasoning Model | | | | | | | |
| DeepSeek-R1-Distill-Llama-8B | None | 39.3 | 27.3 | 84.3 | 88.6 | 36.9 | 55.2 |
| | + DAPO | 38.4 | 26.5 | 86.9 | 89.6 | 38.4 | 55.9 |
| | + $Re^2$ | **47.2** | **29.6** | **88.7** | **92.2** | **44.8** | **60.5** (+4.4) |

parameters are fixed at $\varepsilon_{\text{low}} = 0.2$ and $\varepsilon_{\text{high}} = 0.28$, and the maximum number of redo rounds is $R = 5$. For DAPO, we adopt the same learning rate of $1 \times 10^{-6}$. To ensure comparable token budgets with **$Re^2$**, each batch contains 128 queries with $n = 20$ samples per query. All other hyperparameters are kept identical to those used in $Re^2$. During evaluation, the maximum sequence length is increased to 16384 tokens, with sampling performed using a temperature of 0.6 and top-$p$ of 0.95. For models trained with $Re^2$, whenever a sampled completion produces a *redo* action, sampling is restarted until a final answer is generated, and the first valid final answer is taken as the model's output.

## 5.2 MAIN RESULTS

As shown in Table 1, $Re^2$ improves reasoning performance across all five benchmark datasets and five model types, including base, instruction-tuned, and reasoning-optimized models ranging from 3B to 14B parameters. On pretrained models such as Qwen2.5-7B and Qwen2.5-14B, our method achieves larger gains compared to DAPO. These consistent gains on in-domain mathematical benchmarks of varying difficulty (AIME24, AIME25, AMC, GSM8K) as well as the out-of-domain scientific reasoning benchmark (GPQA-Diamond) demonstrate the robustness of our approach. Moreover, since AIME25 was released after all the evaluated models were trained, it is free from potential data contamination, and $Re^2$ achieves superior performance on this benchmark, further validating its effectiveness. Notably, $Re^2$ achieves substantial improvements on AIME24 and AIME25, highlighting its effectiveness in tackling more challenging reasoning problems.

## 5.3 PERFORMANCE UNDER TEST-TIME SCALING

When tackling challenging problems (e.g., the AIME series), models trained with $Re^2$ may perform multiple redo attempts and generate several candidate solutions before producing a final answer, thereby consuming more tokens during inference. To fairly assess the effect of this additional token usage, we compare DAPO and $Re^2$ under the same number of sampled outputs, regardless of whether a sample corresponds to a direct final answer or a redo attempt. Accuracy is then measured using majority voting over these samples.

As shown in Figure 6, $Re^2$ fully exploits the benefits of test-time scaling: once the number of samples exceeds 64, they consistently surpass RLVR-trained models, whose performance has already saturated, and continue to improve as test-time compute increases. However, when the number

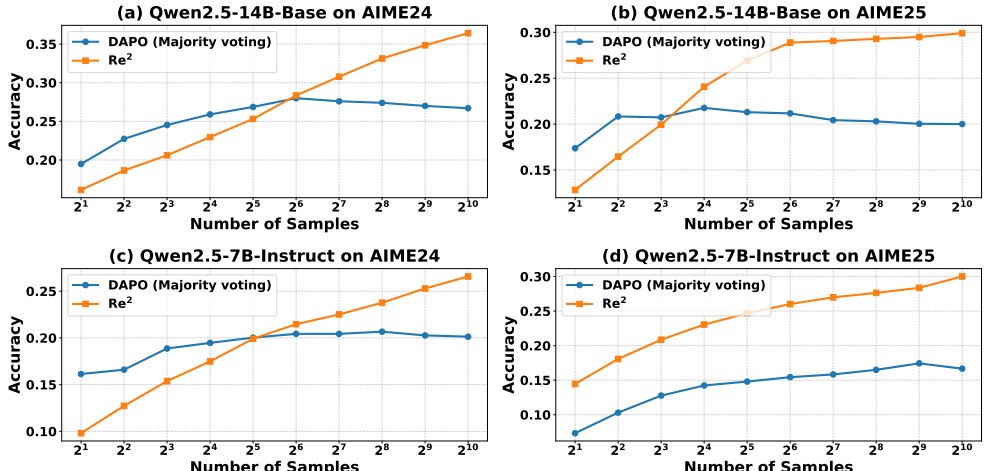

Figure 6: Test-time scaling of DAPO and Re$^2$ under the same training budget as the number of samples increases.

of samples is small, Re$^2$ tends to trigger more redo actions on hard problems, which reduces the proportion of valid final answers within the sampled outputs and can lead to lower accuracy than RLVR under these settings.

## 5.4 Training Dynamics of Re$^2$

Figure 7 shows the training dynamics of Re$^2$, including the average reward, the reward under the resolve action, and the behavioral distribution over correct answers, incorrect answers, and re-solve actions. Both the average reward and the reward for resolving steadily increase as training progresses. In terms of behavior, the probability of choosing to re-solve rises sharply during the first 20 steps and then gradually decreases with further training. Meanwhile, the probability of producing incorrect answers drops substantially, while the probability of generating correct final answers increases slowly. These results suggest that Re$^2$ rapidly activates resolving behavior and then refines it, enabling the model to abandon unproductive reasoning paths while maintaining exploration of correct but more challenging trajectories.

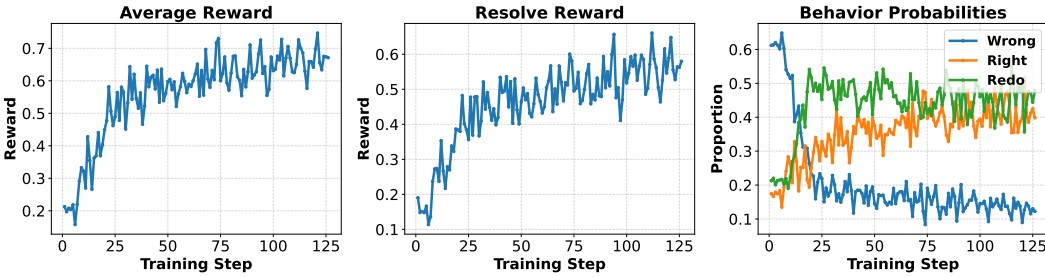

Figure 7: The training progress of Re$^2$ on Qwen2.5-14B-Base.

## 5.5 Analyzing the Mechanisms Behind Re$^2$'s Effectiveness

To further investigate the underlying mechanisms behind the effectiveness of our method, we manually inspected sampled cases and analyzed accuracy improvements across different difficulty levels. Through manual inspection, we found that Re$^2$ produces reasoning chains of consistently higher quality than RLVR algorithms such as DAPO. This is largely because **Re$^2$ allows the model to restart reasoning when the current trajectory is unlikely to lead to the correct answer, thereby substantially reducing the tendency to force a final answer from flawed reasoning or nonexistent conditions.** We believe that the reward modeling in Re$^2$ is fundamentally more rational than the standard 0/1 end-reward paradigm in RLVR. Under a pure end-reward objective, the model is

encouraged to always output a final answer—even when it is uncertain—often producing spurious steps and incoherent logic in an attempt to "guess correctly." In contrast, $Re^2$ enables the model to output a final answer only when it is confident, and to honestly indicate the need to restart when the reasoning becomes unpromising. This more rational and self-aware behavior allows the model to better recognize when its current chain of thought is unreliable and to avoid optimizing trajectories that accidentally guess the correct answer, which is a common issue in RLVR training.

To illustrate these findings, we categorized problems by difficulty and measured the accuracy and resolving rate of $Re^2$. We mixed AMC and AIME25 questions to create a test set with a balanced distribution of difficulty levels. In Fig. 8(a), we grouped questions into seven difficulty levels based on the accuracy of the Qwen2.5-7B-Instruct model obtained through multiple samples. We then evaluated DAPO and $Re^2$ on each difficulty group and recorded $Re^2$'s resolving rate. We observe that for questions the base model is completely unable to solve (Group 1), reinforcement learning cannot teach the model to solve them either. In such cases, DAPO often produces incorrect answers with unclear or erroneous reasoning chains, while $Re^2$ almost always refrains from giving a final answer and attempts to resolve the problem instead. For difficult but solvable questions (Group 2), $Re^2$ achieves more than twice the accuracy of DAPO due to its ability to restart from failed prefixes. Across all difficulty levels, $Re^2$ consistently outperforms DAPO, and its resolving rate decreases as question difficulty decreases.

In Fig. 8(b), we group questions by the difficulty estimated by DAPO, providing a more direct comparison against a standard RLVR method. We find that the largest improvement occurs on questions that RLVR occasionally solves (Group 4), where accuracy increases from 51.2% to 81.7%. These are questions that RLVR models are capable of solving, but their ability to answer correctly is highly unstable because they cannot discard unpromising prefixes. $Re^2$ overcomes this limitation: by allowing the model to restart, it substantially increases the probability of following a successful reasoning trajectory.

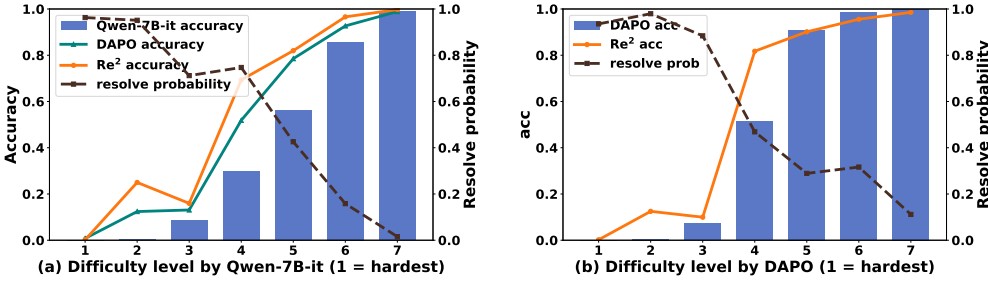

Figure 8: Analysis of accuracy and behavior across problem difficulty levels. **(a)** Accuracy improvements of DAPO and $Re^2$, along with the resolving rate of $Re^2$, when questions are grouped by the difficulty estimated by the base model. **(b)** Accuracy improvements and resolving rate of $Re^2$ when questions are grouped by the difficulty estimated by DAPO.

## 6 CONCLUSION

In this paper, we investigate the limitations of test-time scaling in existing LLMs and show that once early reasoning is misguided, the model rarely recovers and struggles to return to the correct reasoning path. To this end, we propose a new reasoning paradigm that allows language models to flexibly restart reasoning when the current trajectory appears unpromising. We introduce Reinforcement Learning with Re-solving ($Re^2$), which leverages pure reinforcement learning to encourage models to adopt re-solving behaviors. Empirical evaluations demonstrate that $Re^2$ consistently outperforms standard RLVR methods across benchmarks of varying domains and difficulty levels, while also raising the upper bound of performance achievable under test-time scaling.

## ETHICS STATEMENT

This paper presents work aimed at advancing the field of reasoning with large language models. As with all research involving large language models, there are inherent risks, including the spread of misinformation and the propagation of societal biases.

## REPRODUCIBILITY STATEMENT

Our work is based on open-source models and datasets. In Section 5 and Appendix A, we provide detailed descriptions of the prompt templates, method implementation, and experimental setups.

## ACKNOWLEDGMENTS

We want to thank all the anonymous reviewers for their valuable comments. This work was supported by the National Science Foundation of China (NSFC No. 62576232) and the Young Elite Scientists Sponsorship Program by CAST (2023QNRC001).

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

## A  IMPLEMENTATION DETAILS

During training, Re$^2$ employs a specialized template, as illustrated in Figure 9, to elicit redo behavior from the vanilla model. The template is designed to encourage the model to explicitly indicate when it decides to abandon the current reasoning trajectory and restart the problem.

When performing reward assignment, if the model outputs a phrase such as "`It's better to redo the question.`" before producing a boxed final answer, we truncate the response at that point and append a special `<eos>` token to mark the end of the sequence. This ensures that redo actions are clearly distinguished from final-answer completions and allows the policy to learn consistent behavior during reinforcement learning.

---

**Re$^2$ template**

**User:** Solve the following math problem step by step. If you obtain a final answer, enclose it in boxed{}.

{QUESTION}

Note: As you work through the problem, continuously assess your solution path in real time. If you feel your current approach has become unclear or that rethinking the problem from scratch would be more productive, **simply say "It's better to redo the question."** Once you say this, the answer should be immediately terminated (so do not use this phrase lightly—when reflecting to yourself, use "Do I need to redo the question?" instead). When faced with unclear or tangled reasoning, don't hesitate to make use of the option to start over. Use this choice wisely for the best results.

**Assistant:**

---

Figure 9: The template of Re$^2$.

## B  DERIVATION OF THE REWARD OF RE$^2$

We derive the expected reward for choosing `resolve` with at most $R$ allowed rounds. For out-of-group completions (i.e., prefixes $\neq \text{Pre}_i$), let $P_{\neq i}(\texttt{correct})$, $P_{\neq i}(\texttt{incorrect})$, $P_{\neq i}(\texttt{resolve})$

denote the empirical probabilities of the three outcomes, estimated from the $(n-1) \times m$ out-of-group CoTs. Each redo round draws one outcome from this distribution; a `correct` yields reward 1, an `incorrect` yields reward 0, and a `resolve` consumes one round and restarts from the same distribution. Hence, the expected reward of choosing `resolve` equals the probability that the first non-`resolve` outcome within the first $R$ rounds is `correct`:

$$\mathbb{E}[r \mid \texttt{resolve}, R] = \sum_{t=1}^{R} \underbrace{P_{\neq i}(\texttt{resolve})^{t-1}}_{\text{first } t-1 \text{ are } \texttt{resolve}} \cdot \underbrace{P_{\neq i}(\texttt{correct})}_{\text{the } t\text{-th is } \texttt{correct}}$$

$$= P_{\neq i}(\texttt{correct}) \sum_{t=0}^{R-1} P_{\neq i}(\texttt{resolve})^{t}.$$

Evaluating the finite geometric series gives

$$\mathbb{E}[r \mid \texttt{resolve}, R] = P_{\neq i}(\texttt{correct}) \cdot \frac{1 - P_{\neq i}(\texttt{resolve})^{R}}{1 - P_{\neq i}(\texttt{resolve})}.$$

Therefore, for $O_{i,j}$ that chooses `resolve`, we set

$$r_{i,j} = P_{\neq i}(\texttt{correct}) \cdot \frac{1 - P_{\neq i}(\texttt{resolve})^{R}}{1 - P_{\neq i}(\texttt{resolve})}.$$

Table 2: Accuracy with 95% confidence intervals on five reasoning benchmarks, confidence intervals are given in parentheses.

| Models | Methods | AIME24 | AIME25 | AMC23 | GSM8K | GPQA |
|---|---|---|---|---|---|---|
| Base Model | | | | | | |
| Qwen2.5-7B Base | + DAPO | 11.9 (±1.0) | 10.3 (±1.0) | 64.7 (±1.3) | 91.8 (±0.1) | 29.7 (±0.5) |
| | + Re$^2$ | **17.1 (±1.4)** | **19.0 (±1.2)** | **70.8 (±0.3)** | **93.6 (±0.0)** | **36.8 (±0.3)** |
| Qwen2.5-14B Base | + DAPO | 18.2 (±1.2) | 15.7 (±1.2) | 64.0 (±1.3) | 94.3 (±0.1) | 44.8 (±0.6) |
| | + Re$^2$ | **28.5 (±1.1)** | **23.4 (±1.3)** | **68.5 (±0.3)** | **94.6 (±0.0)** | **49.6 (±0.3)** |
| Instruct Model | | | | | | |
| Llama3.2-3B-Instruct | + DAPO | 15.0 (±0.9) | 0.5 (±0.3) | 32.3 (±1.3) | 80.4 (±0.2) | **20.7 (±0.5)** |
| | + Re$^2$ | **17.7 (±1.1)** | **2.8 (±0.5)** | **38.4 (±0.8)** | **83.2 (±0.1)** | 20.2 (±0.3) |
| Qwen2.5-7B-Instruct | + DAPO | 16.0 (±1.1) | 8.6 (±0.9) | 62.3 (±1.3) | 92.6 (±0.1) | 35.4 (±0.6) |
| | + Re$^2$ | **18.6 (±1.6)** | **21.2 (±1.1)** | **64.7 (±0.4)** | **94.1 (±0.0)** | **38.4 (±0.4)** |
| Reasoning Model | | | | | | |
| DeepSeek-R1-Distill-Llama-8B | + DAPO | 38.4 (±1.5) | 26.5 (±1.4) | 86.9 (±0.9) | 89.6 (±0.1) | 38.4 (±0.6) |
| | + Re$^2$ | **47.2 (±0.7)** | **29.6 (±0.8)** | **88.7 (±0.2)** | **92.2 (±0.0)** | **44.8 (±0.3)** |

# C EXPERIMENTS

## C.1 MAIN RESULTS

We additionally provide a comparison between DAPO and Re$^2$ with confidence intervals, as shown in Table 2.

## C.2 PERFORMANCE UNDER TEST-TIME SCALING

We further compare the test-time scaling performance of Re$^2$ with a broader set of baselines. These include: **GRPO** (Shao et al., 2024), the classical RLVR algorithm and the core technique used in training DeepSeek-R1 (Guo et al., 2025); **DLER** (Liu et al., 2025), the recent state-of-the-art efficient reasoning method that reduces token consumption while maintaining performance through truncated-length penalties and training-stabilization strategies; and **DeepConf** (Fu et al., 2025), which leverages internal confidence signals during decoding to dynamically terminate low-quality reasoning traces.

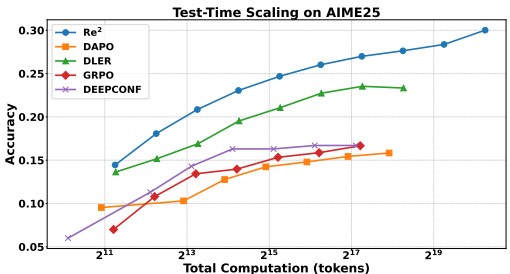

Figure 10: Test-time scaling of Re$^2$ compared with additional baselines.

We use the number of consumed tokens as the measure of computational cost and evaluate them on the challenging AIME25 benchmark, which has no risk of data leakage. As shown in Figure 10, our method achieves better test-time scaling than all competing approaches.

## C.3 ANALYSIS OF THE RESOLVE REWARD ESTIMATOR

To better understand the performance of the resolve reward estimator in Re$^2$, we evaluate how accurately it estimates a model's resolving accuracy. For each training question, we draw 1024 independent samples and treat the proportion of correct responses as the ground-truth resolving accuracy. We then compare the bias and variance of the estimator under different sampling configurations.

Specifically, we fix the number of suffixes at $m = 8$ and vary the number of prefixes $n \in \{2, 4, 8, 16\}$, and conversely fix $n = 8$ while varying $m \in \{2, 4, 8, 16\}$. As a baseline, we include an exponential moving average (EMA) estimator with a decay rate of $0.9$.

As shown in Fig. 11, the $Re^2$ reward estimator becomes increasingly accurate as either $n$ or $m$ increases, exhibiting consistently lower bias and variance. It also outperforms the EMA baseline across all settings, demonstrating the effectiveness of leveraging the naturally generated $n \times m$ suffix samples for estimating resolving accuracy.

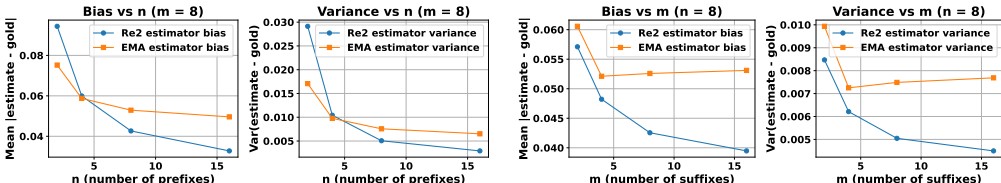

Figure 11: Bias and variance of the estimated resolving accuracy under different values of $n$ and $m$.

## C.4 ANALYSIS OF THE DEGENERATION GROUP RATE

We report the degeneration group rate of DAPO and $Re^2$, defined as the rate of groups in which all samples receive the same reward and therefore have zero advantage. As shown in Fig. 12, in $Re^2$, degeneration groups that are "all-wrong" during the later training stages gradually turn into "all-redo," indicating that the model shifts from forcing an answer on unsolvable questions to choosing to redo them. The overall degeneration rate of $Re^2$ is about 10% higher than that of DAPO. However, as noted in DAPO (Yu et al., 2025), *"the filter strategy does not necessarily impede training efficiency, because the generation time is typically dominated by the generation of long-tail samples if the RL system is synchronized and the generation stage is not pipelined."* Consistent with this observation, $Re^2$ does not incur additional training cost due to this effect.

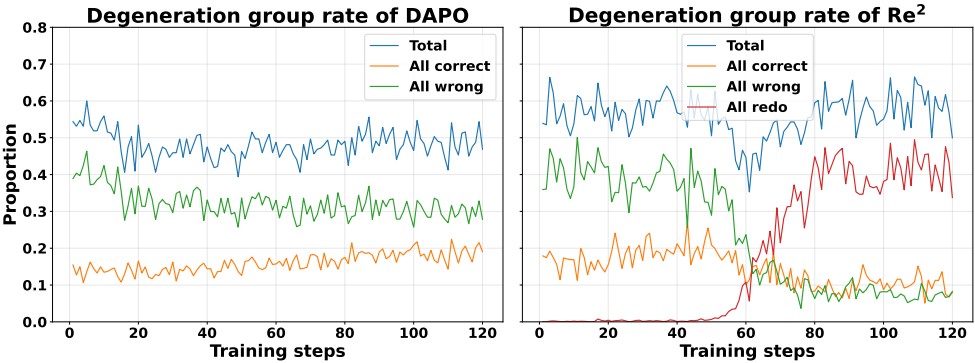

Figure 12: Degeneration group rate during training for DAPO and $Re^2$.

## C.5 ANALYSIS OF THE REASONING LENGTH

We analyze the evolution of reasoning lengths for DAPO and $Re^2$ throughout training. As shown in Figure 13, combined with our inspection of a number of cases, DAPO exhibits steady growth in CoT length during training due to increasing amounts of self-reflection, error correction, and switching of reasoning strategies, similar to classical RLVR. Wrong CoTs are noticeably longer than correct ones because many failed trajectories attempt to extend or patch flawed prefixes. This represents the strategy optimization that occurs under the one-shot chain-of-thought paradigm. In contrast, $Re^2$ gradually stabilizes its CoT lengths, and the lengths of correct, wrong, and redo CoTs do not show substantial differences. Both the statistical results and our qualitative observations indicate that $Re^2$ behaves more rationally under the multi-chain, resolving reasoning paradigm: the model does not force itself into producing a strained and ultimately incorrect reasoning chain, nor does it over-commit to unpromising trajectories.

In addition, redo-CoTs in $Re^2$ tend to be longer at the beginning of training, and their lengths gradually approach those of final correct or incorrect CoTs as training progresses. Consistent with our manual

inspection of cases, early in training, the patterns that trigger a redo are relatively shallow—for example, CoTs becoming excessively long and close to the context window limit, frequent switching between reasoning threads, or resorting to brute-force enumeration on problems that actually require summarizing underlying patterns. In later stages, however, redo decisions become more closely tied to the intrinsic quality of the CoT, such as the effectiveness of the chosen approach or the soundness of the assumptions.

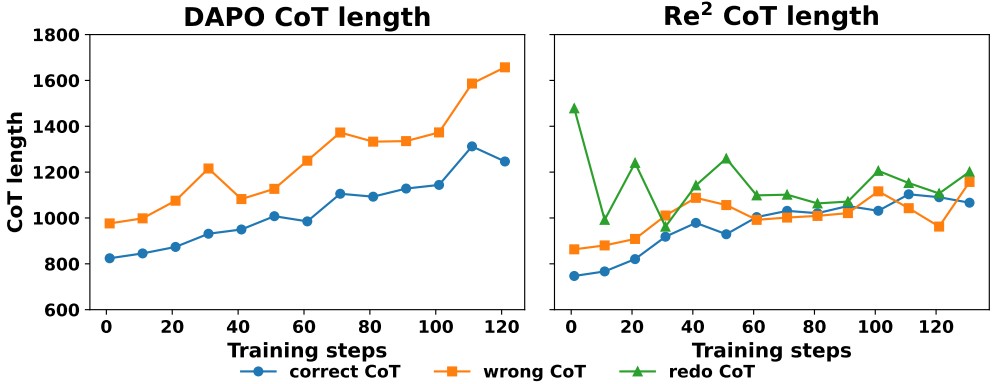

Figure 13: Reasoning lengths during training for DAPO and $Re^2$.

# D TRAINING COST ANALYSIS OF $RE^2$

Training $Re^2$ involves a two-stage generation process: the model first produces $n$ prefixes and then generates $n \times m$ suffixes. This sequential generation procedure can be more time-consuming than directly generating responses in parallel. To quantify this overhead, we measure the rollout time of DAPO and $Re^2$ on an 8×A100-PCIE-40GB server.

To produce the same number of rollout samples, DAPO uses a global batch size of 128, generating 16 samples per question. $Re^2$ uses a global batch size of 32, generating 8 prefixes followed by 8 suffixes for each prefix. On average, DAPO

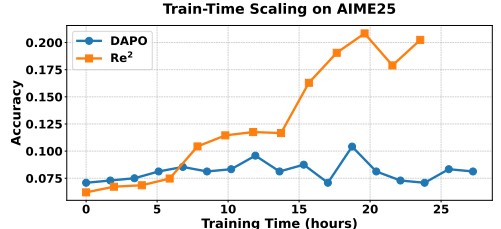

Figure 14: Accuracy improvements of **DAPO** and **$Re^2$** under the same training time.

requires 388 seconds per rollout step. In comparison, $Re^2$ takes 89 seconds for prefix generation and 342 seconds for suffix generation, resulting in a total of 431 seconds, which corresponds to an 11% increase in rollout time.

Considering the performance improvements $Re^2$ provides during training, this additional cost is acceptable. As shown in Fig. 14, $Re^2$ outperforms DAPO under the same training time budget.

# E CASE ANALYSIS OF CoTS WITH DIFFERENT LENGTHS

Here, we present a case study comparing short and long CoT responses obtained from multiple samples on the same problem in Figures 15, 16, 17, and 18. When the model follows a correct reasoning path with high-quality initial steps, it can often reach the correct answer within a relatively short CoT. In contrast, when the initial reasoning direction is misguided, the model tends to produce longer CoTs with many additional tokens, yet still fails to arrive at the correct solution. This illustrates the critical importance of early reasoning quality and highlights the inefficiency of over-extended reasoning when the initial trajectory is flawed.

---

**longer and shorter response in AIME25**

**Problem:** Let $k$ be real numbers such that the system $|25+20i-z| = 5$ and $|z-4-k| = |z-3i-k|$ has exactly one complex solution $z$. The sum of all possible values of $k$ can be written as $\frac{m}{n}$, where $m$ and $n$ are relatively prime positive integers. Find $m + n$. Here $i = \sqrt{-1}$.

**Shorter Response:** Okay, so I need to solve this problem where we have two complex equations:
1. $|25 + 20i - z| = 5$
2. $|z - 4 - k| = |z - 3i - k|$

And the system has exactly one complex solution $z$. We need to find the sum of all possible values of $k$, express it as $m/n$ where $m$ and $n$ are coprime, and then find $m + n$. Alright, let's break this down step by step. First, let's understand what these equations represent geometrically. Since $z$ is a complex number, we can write $z = x + yi$. Then, $25 + 20i - z = 25 + 20i - (x + yi) = (25 - x) + (20 - y)i$. The modulus of this is $\sqrt{(25 - x)^2 + (20 - y)^2} = 5$. So, squaring both sides: $(25 - x)^2 + (20 - y)^2 = 25$. That's the equation of a circle in the complex plane with center at $(25, 20)$ and radius 5. So, $z$ lies on this circle. Now, moving to the second equation: $|z - 4 - k| = |z - 3i - k|$. Let me write $z$ as $x + yi$ again. Then Left side $= |(x + yi) - 4 - k| = |(x - 4 - k) + yi| = \sqrt{(x - 4 - k)^2 + y^2}$, Right side $= |(x + yi) - 3i - k| = |(x - k) + (y - 3)i| = \sqrt{(x - k)^2 + (y - 3)^2}$. Setting them equal: $\sqrt{(x - 4 - k)^2 + y^2} = \sqrt{(x - k)^2 + (y - 3)^2}$. Squaring both sides: $(x - 4 - k)^2 + y^2 = (x - k)^2 + (y - 3)^2$. Let's expand both sides: Left side: $(x - (4 + k))^2 + y^2 = (x - 4 - k)^2 + y^2 = (x^2 - 2(4 + k)x + (4 + k)^2) + y^2$. Right side: $(x - k)^2 + (y - 3)^2 = (x^2 - 2kx + k^2) + (y^2 - 6y + 9)$. Set them equal: $x^2 - 2(4 + k)x + (4 + k)^2 + y^2 = x^2 - 2kx + k^2 + y^2 - 6y + 9$. Simplify both sides by subtracting $x^2$ and $y^2$: $-2(4 + k)x + (4 + k)^2 = -2kx + k^2 - 6y + 9$. Bring all terms to the left: $-2(4 + k)x + (4 + k)^2 + 2kx - k^2 + 6y - 9 = 0$. Simplify term by term: First, combine the $x$ terms: $-2(4 + k)x + 2kx = [-8 - 2k + 2k]x = -8x$. Next, constants: $(4 + k)^2 - k^2 - 9$. Let's compute: $(16 + 8k + k^2) - k^2 - 9 = 16 + 8k + k^2 - k^2 - 9 = 7 + 8k$. So now, equation becomes: $-8x + 7 + 8k + 6y = 0$. So that's the equation of the second condition. So now, we have two conditions: 1. $z$ lies on the circle with center $(25, 20)$ and radius 5; 2. $z$ lies on the line defined by $-8x + 6y + 7 + 8k = 0$. Since the system has exactly one solution, this implies that the line is tangent to the circle. So, the distance from the center of the circle to the line must be equal to the radius. So, the center of the circle is $(25, 20)$, and the line is $-8x + 6y + 7 + 8k = 0$. The distance from the center $(25, 20)$ to the line is equal to the radius 5. The formula for the distance from a point $(x_0, y_0)$ to the line $ax + by + c = 0$ is $|ax_0 + by_0 + c|/\sqrt{a^2 + b^2}$. In our case, the line is $-8x + 6y + (7 + 8k) = 0$, so $a = -8, b = 6, c = 7 + 8k$. The distance is $|-8 * 25 + 6 * 20 + 7 + 8k|/\sqrt{((-8)^2 + 6^2)} = |-200 + 120 + 7 + 8k|/\sqrt{64 + 36} = |(-200 + 120 + 7) + 8k|/\sqrt{100} = |(-73) + 8k|/10$. Set this equal to the radius 5: $|-73 + 8k|/10 = 5 \Rightarrow |8k - 73| = 50$. Therefore, $8k - 73 = 50$ or $8k - 73 = -50$. Solving: Case 1: $8k - 73 = 50 \Rightarrow 8k = 123 \Rightarrow k = 123/8$. Case 2: $8k - 73 = -50 \Rightarrow 8k = 23 \Rightarrow k = 23/8$. Therefore, possible $k$ are $123/8$ and $23/8$. Thus, the sum is $123/8 + 23/8 = (123 + 23)/8 = 146/8 = 73/4$. So $m = 73, n = 4$, which are coprime. Thus, $m + n = 73 + 4 = 77$. **Final Answer** $\boxed{77}$ .(Correct Answer)

Figure 15: Case Analysis of CoTs with Different Lengths in AIME25

---

**longer and shorter response in AIME25**

**Longer Response:** Okay, so I have this problem here where I need to find all real numbers $k$ such that the system of equations:1. $|25 + 20i - z| = 5$ ;2. $|z - 4 - k| = |z - 3i - k|$ has exactly one complex solution $z$. Then, I need to find the sum of all possible values of $k$ and express it as $\frac{m}{n}$ where $m$ and $n$ are coprime, and then find $m + n$.Alright, let's break this down step by step.First, let me understand what each equation represents geometrically.Starting with the first equation: $|25 + 20i - z| = 5$. Since $z$ is a complex number, let's denote $z = x + yi$ where $x$ and $y$ are real numbers.So, substituting $z = x + yi$ into the first equation:$|25 + 20i - (x + yi)| = 5$.Simplify the expression inside the absolute value:$|(25 - x) + (20 - y)i| = 5$ .The modulus of a complex number $a + bi$ is $\sqrt{a^2 + b^2}$, so:$\sqrt{(25 - x)^2 + (20 - y)^2} = 5$.Squaring both sides:$(25 - x)^2 + (20 - y)^2 = 25$.This is the equation of a circle in the complex plane with center at $(25, 20)$ and radius 5.So, the first equation represents a circle centered at $(25, 20)$ with radius 5.Now, moving on to the second equation: $|z - 4 - k| = |z - 3i - k|$.Again, substituting $z = x + yi$:Left side:$|(x - 4 - k) + (y - k)i|$.Right side: $|(x) + (y - 3)i - k|$. Wait, perhaps better to substitute more carefully.Wait, actually, let's note that $z$ is $x + yi$, so:Left side: $|(x + yi) - (4 + k)| = |(x - 4 - k) + yi|$.Right side: $|(x + yi) - (3i + k)| = |(x - k) + (y - 3)i|$.Therefore, the equation becomes:$|(x - 4 - k) + yi| = |(x - k) + (y - 3)i|$.Which is equivalent to:$\sqrt{(x - 4 - k)^2 + y^2} = \sqrt{(x - k)^2 + (y - 3)^2}$.Squaring both sides to remove the square roots:$(x - 4 - k)^2 + y^2 = (x - k)^2 + (y - 3)^2$.Let's expand both sides:Left side: $(x - (4 + k))^2 + y^2 = (x - (4 + k))^2 + y^2$.Which expands to: $x^2 - 2x(4 + k) + (4 + k)^2 + y^2$.Right side: $(x - k)^2 + (y - 3)^2$.Which expands to: $x^2 - 2kx + k^2 + y^2 - 6y + 9$.Now, set left side equal to right side:$x^2 - 2x(4 + k) + (4 + k)^2 + y^2 = x^2 - 2kx + k^2 + y^2 - 6y + 9$.Simplify both sides:First, notice that $x^2$ and $y^2$ appear on both sides, so they cancel out.Left side: $-2x(4 + k) + (4 + k)^2$.Right side: $-2kx + k^2 - 6y + 9$.So, bring all terms to one side:$-2x(4 + k) + (4 + k)^2 - (-2kx) - k^2 + 6y - 9 = 0$.Simplify term by term:First, expand $-2x(4 + k) + 2kx$:Which is $-8x - 2kx + 2kx = -8x$. Wait, perhaps let's do it step by step:After cancelling $x^2$ and $y^2$, we have:Left:$-2x(4 + k) + (4 + k)^2$.Right: $-2kx + k^2 - 6y + 9$.So, subtract the right side from both sides:$-2x(4 + k) + (4 + k)^2 - (-2kx) - k^2 + 6y - 9 = 0$.Simplify each term:First term: $-2x(4 + k) + 2kx$.Second term: $(4 + k)^2 - k^2$.Third term: $6y - 9$.Compute each part:1. $-2x(4 + k) + 2kx = (-8x - 2kx) + 2kx = -8x$,2.$(4 + k)^2 - k^2 = (16 + 8k + k^2) - k^2 = 16 + 8k$,3.$6y - 9$.(Error 1)**Putting it all together:**$-8x + 16 + 8k + 6y - 9 = 0$.**Simplify constants:**$16 - 9 = 7$, **so:**$-8x + 8k + 7 + 6y = 0$.**Let's rewrite:**$-8x + 6y + (8k + 7) = 0$,**We can divide the entire equation by -1 to make it look cleaner:**$8x - 6y - (8k + 7) = 0$.**Alternatively, keep it as:**$-8x + 6y + 8k + 7 = 0$.This is the equation representing the set of points $z = x + yi$ such that$|z - 4 - k| = |z - 3i - k|$.This equation is actually the perpendicular bisector of the segment joining the points $4 + k$ and $3i + k$ in the complex plane.Wait, let's think in terms of coordinates.(Error 2)**In the complex plane, $z$ is a point**$(x, y)$**. The equation** $|z - (4 + k)| = |z - (3i + k)|$**can be rewritten as** $|(x + yi) - (4 + k)| = |(x + yi) - (0 + (3 + k)i)|$.**So, this is the set of points equidistant from** $(4, k)$ **and**$(0, 3 + k)$. Therefore, the perpendicular bisector of the segment connecting $(4, k)$ and $(0, 3 + k)$.Therefore, the second equation represents the perpendicular bisector of the segment between $(4, k)$ and $(0, 3 + k)$.So, now we have two conditions:1. $z$ lies on the circle centered at $(25, 20)$with radius 5;2.$z$ lies on the perpendicular bisector of the segment between $(4, k)$ and $(0, 3 + k)$.We are told that the system has exactly one solution. That means that the line (perpendicular bisector) intersects the circle at exactly one point. Therefore, the line must be tangent to the circle.Therefore, the distance from the center of the circle to the line must be equal to the radius of the circle.So, perhaps the steps are:1. Find the equation of the perpendicular bisector of the segment between $(4, k)$ and $(0, 3 + k)$.2. Find the distance from the center of the circle $(25, 20)$ to this line.3. Set this distance equal to the radius 5, and solve for $k$.4. Sum all possible values of $k$.So, let's proceed step by step.First, let's find the equation of the perpendicular bisector.Given two points $A(4, k)$ and $B(0, 3 + k)$.The midpoint $M$ of segment AB is:$M_x = \frac{4 + 0}{2} = 2, M_y = \frac{k + (3 + k)}{2} = \frac{3 + 2k}{2}$So, midpoint is $(2, (3 + 2k)/2)$.Next, the slope of AB is:$m_{AB} = \frac{(3 + k) - k}{0 - 4} = \frac{3}{-4} = -3/4$.Therefore, the slope of the perpendicular bisector is the negative reciprocal, which is $4/3$.Therefore, the equation of the perpendicular bisector is:$y - M_y = (4/3)(x - M_x)$.Plugging in $M_x = 2$and $M_y = (3 + 2k)/2$:$y - (3 + 2k)/2 = (4/3)(x - 2)$.

Figure 16: Case Analysis of CoTs with Different Lengths in AIME25

---

**longer and shorter response in AIME25**

Let's write this in standard form $ax + by + c = 0$.Multiply both sides by 3 to eliminate the fraction:$3(y - (3 + 2k)/2) = 4(x - 2)$.Simplify:$3y - (9 + 6k)/2 = 4x - 8$.Multiply all terms by 2 to eliminate denominator:$6y - 9 - 6k = 8x - 16$.Bring all terms to left side: $-8x + 6y - 9 - 6k + 16 = 0$.Simplify constants:$-9 + 16 = 7$.Thus:$-8x + 6y + 7 - 6k = 0$ Or, $-8x + 6y + (7 - 6k) = 0$ .We can write this as: $8x - 6y + (6k - 7) = 0$ .Wait, let me check.From above:After multiplying by 2:$6y - 9 - 6k = 8x - 16$.Then, moving all terms to left:$-8x + 6y + (-9 - 6k + 16) = 0$.Which is:$n - 8x + 6y + (7 - 6k) = 0$.So, if we want to write it as $ax + by + c = 0$, (Error 1)**then:$-8x + 6y + (7 - 6k) = 0$.Alternatively, multiply both sides by -1:$8x - 6y - 7 + 6k = 0$,Which is:$8x - 6y + 6k - 7 = 0$.Thus, the equation is** $8x - 6y + (6k - 7) = 0$.**Alternatively, let's verify.Alternatively, perhaps it's better to keep the equation as:From above: -$8x + 6y + 7 - 6k = 0$ .Which can be written as:$8x - 6y = 7 - 6k$.But perhaps let's consider the standard form as:$8x - 6y + (-7 + 6k) = 0$.Either way, perhaps it's better to find the distance from the center $(25, 20)$ to this line.The distance from a point $(x_0, y_0)$ to the line $ax + by + c = 0$ is $|ax_0 + by_0 + c|/\sqrt{(a^2 + b^2)}$.So, first, let's write the equation in standard form.From above, the equation is:$-8x + 6y + 7 - 6k = 0$.We can write it as:$-8x + 6y + (7 - 6k) = 0$.Thus, coefficients are $a = -8, b = 6, c = 7 - 6k$.Therefore, distance from $(25, 20)$ is:$|-8*25 + 6*20 + (7 - 6k)|/\sqrt{((-8)^2 + 6^2)}$.Compute numerator:First compute each term:$-8 * 25 = -200, 6 * 20 = 120$.So, total so far: $-200 + 120 = -80$.Add $(7 - 6k)$: $-80 + 7 - 6k = -73 - 6k$.So numerator is $|-73 - 6k|$.Denominator is $\sqrt{64 + 36} = \sqrt{100} = 10$.Since the distance must be equal to radius 5, we have:$|-73 - 6k|/10 = 5$.Therefore,$|-73 - 6k| = 50$.Which implies:$-73 - 6k = 50$ or $-73 - 6k = -50$Case 1: $-73 - 6k = 50, -6k = 50 + 73 = 123, k = -123/6 = -41/2$.Case 2: $-73 - 6k = -50, -6k = -50 + 73 = 23, k = -23/6$.Therefore, the possible values of $k$ are $-41/2$ and $-23/6$.Therefore, the sum of all possible values is:$-41/2 + -23/6 = $ convert to common denominator:$-41/2 = -123/6$, so total is $-123/6 - 23/6 = (-146)/6 = -73/3$.(Error 3)**But wait, the question says "the sum of all possible values of $k$" so perhaps we need to check if both solutions are valid**.Wait, perhaps let's see if these $k$'s result in exactly one solution.Since each $k$ gives a different line, each line can intersect the circle at two points, one point, or none. Since we have exactly one solution, it's when the line is tangent to the circle.Therefore, perhaps both $k$'s are valid, but perhaps we need to check if both lead to lines that are tangent to the circle.Alternatively, perhaps only one of them is valid because for some $k$, the line might not intersect the circle or something. But perhaps both are valid.Therefore, perhaps sum is $(-41/2) + (-23/6)=$ as above, which is $(-123/6 - 23/6) = (-146)/6 = (-73)/3$.But since the problem says "the sum of all possible values of $k$", which could be negative or positive,**but perhaps we need to check if both are valid**.Alternatively, perhaps the sum is (-41/2) + (-23/6) = (-123 -23)/6= (-146)/6= -73/3.**But since the problem says "sum of all possible values of $k$", perhaps we can write it as a fraction.But perhaps let's verify**.First, let's check for $k = -41/2$.Compute the distance from center to the line: $|-73 - 6 * (-41/2)|/10$.Compute:$-73 - 6 * (-41/2) = -73 + (6 * 41)/2 = -73 + (246)/2 = -73 + 123 = 50$.Thus, $|50|/10 = 5$, which is the radius.So this is correct.Similarly, for $k = -23/6$:Compute $|-73 - 6 * (-23/6)| = |-73 + 23| = |-50| = 50$.Thus, same result, distance is 5.Therefore, both $k$'s are valid.Hence, the sum is $(-41/2) + (-23/6) = (-123/6 - 23/6) = (-146)/6 = (-73)/3$.Thus, $m = -73, n = 3$. Wait, but $m$ and $n$ must be positive integers. So perhaps take absolute values? **Wait, perhaps the problem expects the sum as positive.But perhaps not, since k can be negative. The problem says "sum of all possible values of $k$", so perhaps we just add them as they are**.Thus, $(-41/2) + (-23/6) = (-41 * 3 - 23)/6 = (-123 - 23)/6 = (-146)/6 = (-73)/3$.So, the sum is $-73/3$, which can be written as $-73/3$. Since $m$ and $n$ are coprime, $m = 73, n = 3$, **but since the sum is negative, perhaps the question expects the absolute value? But no, because the problem says "the sum of all possible values of $k$ can be written as $m/n$"**. ......Therefore, the sum is $-73/3$, so $m = 73, n = 3$, which are coprime. Thus, $m + n = 73 + 3 = 76$.**Final Answer** $\boxed{76}$ (Wrong Answer)

(**Error:**1.Repeated reformulation of the equation form. 2.Coordinate extraction error,renders all following calculations invalid. 3.Ineffective repeated verification of the answer)

Figure 17: Case Analysis of CoTs with Different Lengths in AIME25

---

**longer and shorter response in AMC**

**Problem:** How many positive perfect squares less than 2023 are divisible by 5?

**Shorter Response:** To determine how many positive perfect squares less than 2023 are divisible by 5, we start by noting that a perfect square is divisible by 5 if and only if the integer itself is divisible by 5. This is because if $n^2$ is divisible by 5, then $n$ must be divisible by 5 (since 5 is a prime number). Let $n = 5k$ where $k$ is an integer. Then, $n^2 = (5k)^2 = 25k^2$. We need $25k^2 < 2023$. Solving for $k$, we get: $25k^2 < 2023 \implies k^2 < \frac{2023}{25} \implies k^2 < 80.92$. The largest integer $k$ that satisfies this inequality is 8, because $8^2 = 64$ and $9^2 = 81$ which is greater than 80.92. Therefore, $k$ can take any integer value from 1 to 8. This gives us 8 possible values for $k$. Thus, there are 8 positive perfect squares less than 2023 that are divisible by 5. The answer is: $\boxed{8}$ (Correct Answer)

**Longer Response:** To determine how many positive perfect squares less than 2023 are divisible by 5, we start by noting that a perfect square divisible by 5 must itself be a multiple of 25 (since the square of any integer that is not a multiple of 5 cannot be divisible by 5, and the square of an integer that is a multiple of 5 must be a multiple of $5^2 = 25$. We need to find the largest integer $n$ such that $25n^2 < 2023$. This is equivalent to finding the largest integer $n$ such that $n^2 < \frac{2023}{25} = 80.92$. Since $n$ must be an integer, we take the greatest integer less than $\sqrt{80.92}$, which is 8 (since $\sqrt{80.92} \approx 8.996$). Thus, $n$ can take any integer value from 1 to 8. (Error)**We can verify this by calculating the squares of these integers and checking if they are less than 2023**: $1^2 = 1, 2^2 = 4, 3^2 = 9, 4^2 = 16, 5^2 = 25, 6^2 = 36, 7^2 = 49, 8^2 = 64, 9^2 = 81, 10^2 = 100, 11^2 = 121, 12^2 = 144, 13^2 = 169, 14^2 = 196, 15^2 = 225, 16^2 = 256, 17^2 = 289, 18^2 = 324, 19^2 = 361, 20^2 = 400, 21^2 = 441, 22^2 = 484, 23^2 = 529, 24^2 = 576, 25^2 = 625, 26^2 = 676, 27^2 = 729, 28^2 = 784, 29^2 = 841, 30^2 = 900, 31^2 = 961, 32^2 = 1024, 33^2 = 1089, 34^2 = 1156, 35^2 = 1225, 36^2 = 1296, 37^2 = 1369, 38^2 = 1444, 39^2 = 1521, 40^2 = 1600, 41^2 = 1681, 42^2 = 1764, 43^2 = 1849, 44^2 = 1936, 45^2 = 2025$. Since $45^2 = 2025$ is not less than 2023, the largest integer $n$ is 44. Therefore, the positive perfect squares less than 2023 that are divisible by 5 are the squares of the integers from 1 to 44, which are 44 in total. Thus, the number of positive perfect squares less than 2023 that are divisible by 5 is: $\boxed{44}$. (Wrong Answer) (**Error:** Overlooking the "divisible by 5" constraint during verification.)

Figure 18: Case Analysis of CoTs with Different Lengths in AMC

## F  RE$^2$ EXAMPLES

We present examples of **Re**$^2$ in Figures 19, 20, 21, and 22.

---

**Re$^2$ Examples 1**

**Problem:**How many ordered pairs of positive real numbers $(a, b)$ satisfy the equation$(1 + 2a)(2 + 2b)(2a + b) = 32ab$?

**Reasoning Trace of Re$^2$:**To solve the equation $(1 + 2a)(2 + 2b)(2a + b) = 32ab$ for ordered pairs $(a, b)$ of positive real numbers, we start by simplifying and analyzing the equation.First, let's rewrite the equation for clarity:$(1 + 2a)(2 + 2b)(2a + b) = 32ab$.We can factor out the common terms in the first two factors:$(1 + 2a)2(1 + b)(2a + b) = 32ab$.This simplifies to:$2(1 + 2a)(1 + b)(2a + b) = 32ab$.Dividing both sides by 2, we get:$(1 + 2a)(1 + b)(2a + b) = 16ab$.Next, we will test some simple values for $a$ and $b$ to see if we can find any solutions. Let's start with $a = b$. Substituting $a = b$ into the equation, we get:$(1 + 2a)(1 + a)(2a + a) = 16a^2$.This simplifies to:$(1 + 2a)(1 + a)(3a) = 16a^2$.Expanding the left side, we get:$3a(1 + 2a)(1 + a) = 3a(1 + 3a + 2a^2) = 3a + 9a^2 + 6a^3$.So the equation becomes:$3a + 9a^2 + 6a^3 = 16a^2$.Rearranging terms, we get:$6a^3 + 9a^2 + 3a - 16a^2 = 0$,which simplifies to:$6a^3 - 7a^2 + 3a = 0$.Factoring out $a$, we get:$a(6a^2 - 7a + 3) = 0$.Since $a$ is a positive real number, $a \neq 0$. Therefore, we need to solve the quadratic equation:$6a^2 - 7a + 3 = 0$.The discriminant of this quadratic equation is:$(-7)^2 - 4 \cdot 6 \cdot 3 = 49 - 72 = -23$.Since the discriminant is negative, the quadratic equation has no real solutions.
Therefore, there are no positive real solutions for $a$ when $a = b$. It's better to redo the question.
**Restarting from scratch:**
Next, let's try another approach by assuming $a = \frac{1}{2}$ and solving for $b$:$(1+2\cdot\frac{1}{2})(2+2b)(2\cdot\frac{1}{2}+b) = 32 \cdot \frac{1}{2} \cdot b$,which simplifies to:$(1+1)(2+2b)(1+b) = 16b$ or $2(2+2b)(1+b) = 16b$. Simplifying further, we get:$2(2 + 2b + 2b + 2b^2) = 16b$ or $2(2 + 4b + 2b^2) = 16b$,which simplifies to:$4 + 8b + 4b^2 = 16b$ or $4b^2 - 8b + 4 = 0$.Dividing by 4, we get:$b^2 - 2b + 1 = 0$,which factors to:$(b - 1)^2 = 0$.So, $b = 1$. Therefore, $(a, b) = (\frac{1}{2}, 1)$ is a solution.To confirm that there are no other solutions, we can use the symmetry and the nature of the equation. Given the complexity and the fact that we have found a solution, we can conclude that there are no other positive real solutions.Thus, the number of ordered pairs $(a, b)$ that satisfy the equation is: $\boxed{1}$.(Correct Answer)

---

Figure 19: Re$^2$ Examples 1

## G  LIMITATIONS AND FUTURE WORK

In this section, we outline several limitations of our study that highlight areas for future research and improvement: **(1)** Although our method improves performance on challenging problems through better test-time scaling, models may still require multiple re-solving rounds to reach the correct answer. At present, there is no mechanism to explicitly control the probability of invoking the re-solve action during inference, which may affect efficiency. **(2)** Our experiments primarily focus on large language models, leaving the potential of applying Re$^2$ to other generative modalities (e.g., vision or multi-modal reasoning) unexplored. **(3)** We mainly study reasoning benchmarks, while extending Re$^2$ to more complex tasks, such as tool use, or search-intensive problems could also provide benefits.

## H  RE$^2$ APPLY IN PANGU

OpenPangu-Embedded-1B is a fast-thinking language model trained from scratch on Ascend NPUs and designed for efficient execution on edge devices. We performed post-training on eight A100 GPUs using our methods, with training details consistent with 5.1. As shown in Table3, the accuracy of our method on the four benchmarks is significantly better than the original model. The accuracy improvement on GSM8K even reached 16.6%. This demonstrates that our method remains effective on models designed for efficient execution on edge devices trained with NPUs.

---

**Re$^2$ Examples 2**

**Problem:**Ralph has a record store where people can trade their own records for new ones. People can trade 2 old records for 1 new one. 5 people come in with old records and leave with 7 new records between them. How many old records did the 5 people bring in?

**Reasoning Trace of Re$^2$:**To determine how many old records the 5 people brought to Ralph's record store, we need to understand the trade ratio and the total number of new records they received.The trade ratio is 2 old records for 1 new record. So, if each person trades 2 old records for 1 new record, and 5 people are trading, we need to find out how many old records were traded in total to get 7 new records.Let's denote the number of new records each person receives as $n$. Since each new record is obtained by trading 2 old records, if each of the 5 people receives $n$ new records, the total number of new records is $5n$. We know that the total number of new records they leave with is 7, so we have: $5n = 7$ .Solving for $n$ :$n = \frac{7}{5}$.This means each person receives $\frac{7}{5}$ new records.

However, since each person can only receive a whole number of new records, we need to consider the total number of new records and the trade ratio in another way. It's better to redo the question.
**Restarting from scratch:**
If 5 people together leave with 7 new records, and each new record is obtained by trading 2 old records, the total number of old records traded is: 7 new records $\times$ 2 old records per new record $=$ 14 old records .Therefore, the 5 people brought in a total of 14 old records. So, the number of old records the 5 people brought in is: $\boxed{14}$ (Correct Answer)

---

Figure 20: Re$^2$ Examples 2

Table 3: The accuracy of OpenPangu-Embedded-1B and Re$^2$ on four benchmarks.

| Methods | AIME25 | AMC23 | MATH500 | GSM8K |
|---|---|---|---|---|
| OpenPangu-Embedded-1B | 4.7 | 22.2 | 49.3 | 64.4 |
| + Re$^2$ | **8.6** | **35.3** | **65.8** | **81.0** |

## I   USE OF LLMS

LLMs were used solely for language polishing of the manuscript.

---

**Re$^2$ Examples 3**

**Problem:**Let $x, y$ and $z$ be positive real numbers that satisfy the following system of equations:$\log_2(\frac{x}{yz} = \frac{1}{2}\log_2(\frac{y}{xz}) = \frac{1}{3}\log_2(\frac{z}{xy}) = \frac{1}{4}$.Then the value of $\left|\log_2(x^4y^3z^2)\right|$ is $\frac{m}{n}$ where $m$ and $n$ are relatively prime positive integers. Find $m + n$.

**Reansoning Trace of Re$^2$:**Let's start by expressing the given logarithmic equations in terms of $x$, $y$, and $z$.The given equations are:$\log_2\left(\frac{x}{yz}\right) = \frac{1}{2}$,$\log_2\left(\frac{y}{xz}\right) = \frac{1}{3}$,$\log_2\left(\frac{z}{xy}\right) = \frac{1}{4}$.We can rewrite these equations as:$\log_2(x) - \log_2(yz) = \frac{1}{2}$,$\log_2(y) - \log_2(xz) = \frac{1}{3}$,$\log_2(z) - \log_2(xy) = \frac{1}{4}$.Using the properties of logarithms, we can further simplify these equations:$\log_2(x) - (\log_2(y) + \log_2(z)) = \frac{1}{2}$,$\log_2(y) - (\log_2(x) + \log_2(z)) = \frac{1}{3}$,$\log_2(z) - (\log_2(x) + \log_2(y)) = \frac{1}{4}$.Let's denote:$\log_2(x) = a$, $\log_2(y) = b$, $\log_2(z) = c$.So the equations become:$a - (b+c) = \frac{1}{2}$,$b - (a+c) = \frac{1}{3}$,$c - (a+b) = \frac{1}{4}$.Simplifying each equation, we get:$a - b - c = \frac{1}{2}$ (Equation 1),$b - a - c = \frac{1}{3}$ (Equation 2),$c - a - b = \frac{1}{4}$ (Equation 3).We now have a system of linear equations:$a - b - c = \frac{1}{2}$ (Equation 1),$-b + a - c = \frac{1}{3}$ (Equation 2),$-c - a + b = \frac{1}{4}$ (Equation 3).Let's add all three equations together:$(a - b - c) + (-b + a - c) + (-a - b + c) = \frac{1}{2} + \frac{1}{3} + \frac{1}{4}$.Simplifying the left side:$a - b - c - b + a - c - a - b + c = 2a - 3b - 2c = \frac{1}{2} + \frac{1}{3} + \frac{1}{4}$.The right side can be simplified as:$\frac{1}{2} + \frac{1}{3} + \frac{1}{4} = \frac{6}{12} + \frac{4}{12} + \frac{3}{12} = \frac{13}{12}$.So we have:$2a - 3b - 2c = \frac{13}{12}$.However, let's instead solve for $a$, $b$, and $c$ by subtracting pairs of equations to find individual expressions.Subtract Equation 2 from Equation 1:$(a - b - c) - (-b + a - c) = \frac{1}{2} - \frac{1}{3}$,$a - b - c + b - a + c = \frac{1}{2} - \frac{1}{3}$,$0 = \frac{1}{2} - \frac{1}{3} = \frac{3}{6} - \frac{2}{6} = \frac{1}{6}$.This simplifies to:$0 = \frac{1}{6}$.This is a contradiction, which indicates we need to recheck our method of solving. Instead, let's solve for $a$, $b$, and $c$ by another method.It's better to redo the question.
**Restarting from scratch:**
Let's add Equation 1, Equation 2, and Equation 3 and then solve for $a, b$, and $c$ individually.From the system:$a - b - c = \frac{1}{2}$ (Equation 1),$b - a - c = \frac{1}{3}$ (Equation 2),$c - a - b = \frac{1}{4}$ (Equation 3).We can add these equations:$(a - b - c) + (b - a - c) + (c - a - b) = \frac{1}{2} + \frac{1}{3} + \frac{1}{4}$,$a - b - c + b - a - c + c - a - b = \frac{13}{12}$,$-a - c - b = \frac{13}{12}$,$-(a + b + c) = \frac{13}{12}$,$a + b + c = -\frac{13}{12}$.This is incorrect, so let's re-evaluate by solving the system of linear equations.Instead, let's solve for $a$, $b$, and $c$ directly. It's better to redo the question.
**Restarting from scratch:**
We can use a different approach by expressing $a$, $b$, and $c$ in terms of a common variable. Let's add and subtract the equations in a different way.From:$a - b - c = \frac{1}{2}, b - a - c = \frac{1}{3}, c - a - b = \frac{1}{4}$.Let's solve these equations by eliminating variables step-by-step. We can start by expressing $c$ in terms of $a$ and $b$ from the first equation:$c = a - b - \frac{1}{2}$.Substitute $c$ into the second equation:$b - a - (a - b - \frac{1}{2}) = \frac{1}{3}$,$b - a - a + b + \frac{1}{2} = \frac{1}{3}$,$2b - 2a + \frac{1}{2} = \frac{1}{3}$,$2(b - a) = \frac{1}{3} - \frac{1}{2} = \frac{2}{6} - \frac{3}{6} = -\frac{1}{6}$,$2(b - a) = -\frac{1}{6}$,$b - a = -\frac{1}{12}$,$b = a - \frac{1}{12}$.Now substitute $b = a - \frac{1}{12}$ and $c = a - b - \frac{1}{2}$ into the third equation:$c = a - (a - \frac{1}{12}) - \frac{1}{2} = a - a + \frac{1}{12} - \frac{1}{2} = \frac{1}{12} - \frac{1}{2} = \frac{1}{12} - \frac{6}{12} = -\frac{5}{12}$.Substitute $b = a - \frac{1}{12}$ and $c = -\frac{5}{12}$ into the third equation:$-\frac{5}{12} - a - (a - \frac{1}{12}) = \frac{1}{4}$,$-\frac{5}{12} - a - a + \frac{1}{12} = \frac{1}{4}$,$-\frac{5}{12} + \frac{1}{12} - 2a = \frac{1}{4}$,$-\frac{4}{12} - 2a = \frac{1}{4}$,$-\frac{1}{3} - 2a = \frac{1}{4}$,$-2a = \frac{1}{4} + \frac{1}{3} = \frac{3}{12} + \frac{4}{12} = \frac{7}{12}$.$-2a = \frac{7}{12}, a = -\frac{7}{24}$.Substitute $a = -\frac{7}{24}$ back to find $b$ and $c$:$b = a - \frac{1}{12} = -\frac{7}{24} - \frac{1}{12} = -\frac{7}{24} - \frac{2}{24} = -\frac{9}{24} = -\frac{3}{8}$.$c = -\frac{5}{12}$.Now we have:$a = -\frac{7}{24}$, $b = -\frac{3}{8}$, $c = -\frac{5}{12}$.We need to find $\left(\left|\log_2(x^4y^3z^2)\right|\right)$. First, we express $x^4y^3z^2$ in terms of $a$, $b$, and $c$:$\log_2(x^4y^3z^2) = 4a + 3b + 2c$.Substituting $a$, $b$, and $c$ : $4a + 3b + 2c = 4\left(-\frac{7}{24}\right) + 3\left(-\frac{3}{8}\right) + 2\left(-\frac{5}{12}\right) = -\frac{28}{24} - \frac{9}{8} - \frac{10}{12} = -\frac{28}{24} - \frac{27}{24} - \frac{20}{24} = -\frac{28+27+20}{24} = -\frac{75}{24} = -\frac{25}{8}$.Thus,$\left|\log_2(x^4y^3z^2)\right| = \left|-\frac{25}{8}\right| = \frac{25}{8}$.So, $\left|\log_2(x^4y^3z^2)\right| = \frac{25}{8}$. The fraction $\frac{25}{8}$ is in simplest form, so $m = 25$ and $n = 8$. Therefore, $m + n = 25 + 8 = 33$.The final answer is: $\boxed{33}$ (Correct Answer)

Figure 21: Re$^2$ Examples 3

---

**Re$^2$ Examples 4**

**Problem:** Let $f$ be the unique function defined on the positive integers such that $\sum_{d|n} d \cdot f\left(\frac{n}{d}\right) = 1$ for all positive integers $n$. What is $f(2023)$?

**Reansoning Trace of Re$^2$:** To determine the function $f$ defined on the positive integers such that $\sum_{d|n} d \cdot f\left(\frac{n}{d}\right) = 1$ .for all positive integers $n$ , we start by analyzing the functional equation. We will use the properties of the function and the structure of the equation to find $f(n)$ .Step-by-Step Analysis
1. Base Case: $n = 1$ ,$\sum_{d|1} d \cdot f\left(\frac{1}{d}\right) = 1$. The only divisor of 1 is 1 itself, so the sum reduces to: $1 \cdot f(1) = 1 \implies f(1) = 1$.
2. General Case: $n = p$ where $p$ is a prime number,$\sum_{d|p} d \cdot f\left(\frac{p}{d}\right) = 1$.The divisors of $p$ are 1 and $p$, so the sum becomes: $1 \cdot f(p) + p \cdot f(1) = 1 \implies f(p) + p \cdot 1 = 1 \implies f(p) + p = 1 \implies f(p) = 1 - p$.
3. General Case: $n = p_1^{e_1} p_2^{e_2} \cdots p_k^{e_k}$ (where $p_i$ are distinct primes,$\sum_{d|n} d \cdot f\left(\frac{n}{d}\right) = 1$. For $n = p_1^{e_1}$ , we have:$\sum_{d|p_1^{e_1}} d \cdot f\left(\frac{p_1^{e_1}}{d}\right) = 1$. The divisors of $p_1^{e_1}$ are $(1, p_1, p_1^2, \ldots, p_1^{e_1})$. So, the sum becomes:$\sum_{j=0}^{e_1} p_1^j \cdot f(p_1^{e_1-j}) = 1$.Substituting $f(p_1^m) = 1 - p_1^m$ for $m \geq 1$ and $f(1) = 1$ , we get:$p_1^0 \cdot f(p_1^{e_1}) + p_1^1 \cdot f(p_1^{e_1-1}) + p_1^2 \cdot f(p_1^{e_1-2}) + \cdots + p_1^{e_1} \cdot f(1) = 1 \implies (1 - p_1^{e_1}) + p_1 \cdot (1 - p_1^{e_1-1}) + p_1^2 \cdot (1 - p_1^{e_1-2}) + \cdots + p_1^{e_1} \cdot 1 = 1 \implies (1 - p_1^{e_1}) + p_1(1 - p_1^{e_1-1}) + p_1^2(1 - p_1^{e_1-2}) + \cdots + p_1^{e_1} = 1 \implies (1 - p_1^{e_1}) + p_1 - p_1^{e_1} + p_1^2 - p_1^{e_1} + \cdots + p_1^{e_1} = 1.(1 + p_1 + p_1^2 + \cdots + p_1^{e_1}) - (p_1^{e_1} + p_1^{e_1} + \cdots + p_1^{e_1}) = 1 \implies (1 + p_1 + p_1^2 + \cdots + p_1^{e_1}) - p_1^{e_1} \cdot e_1 = 1$. The sum $1 + p_1 + p_1^2 + \cdots + p_1^{e_1}$ is a geometric series:$(p_1^{e_1+1} - 1)/(p_1 - 1)$.So we get:$(p_1^{e_1+1} - 1)/(p_1 - 1) - p_1^{e_1} \cdot e_1 = 1$. Simplifying this, we need to check if this holds for general $n$ . It's better to redo the question.
Restarting from scratch:
Let's hypothesize that $f(n) = \frac{1}{n}$. We substitute $f(n) = \frac{1}{n}$ into the original equation:
$\sum_{d|n} d \cdot f\left(\frac{n}{d}\right) = \sum_{d|n} d \cdot \frac{1}{\frac{n}{d}} = \sum_{d|n} d \cdot \frac{d}{n} = \sum_{d|n} \frac{d^2}{n} = \frac{1}{n} \sum_{d|n} d^2$.For this to equal 1, we need:$\frac{1}{n} \sum_{d|n} d^2 = 1 \implies \sum_{d|n} d^2 = n$.This is not generally true for all $n$ , so we need to re-evaluate our hypothesis. It's better to redo the question.
Restarting from scratch:
Instead, let's directly use the functional equation and the values we have derived.
Determining $f(2023)$.First, factorize 2023:$2023 = 7 \times 17^2$.Using the functional equation and the derived values, we need to find $f(2023)$ . We can use the multiplicative property of $f$ if we assume $f$ is multiplicative. That means if $n = p_1^{e_1} p_2^{e_2} \cdots p_k^{e_k}$, then $f(n) = f(p_1^{e_1}) f(p_2^{e_2}) \cdots f(p_k^{e_k})$.We already have:$f(p) = 1 - p$ for prime $p$.So:$f(7) = 1 - 7 = -6, f(17) = 1 - 17 = -16$.Since $2023 = 7 \times 17^2$, we have:$f(2023) = f(7) \cdot f(17^2)$.We need to determine $f(17^2)$ : $\sum_{d|17^2} d \cdot f\left(\frac{17^2}{d}\right) = 1$.The divisors of $17^2$ are $(1, 17, 17^2)$. So the sum is:$1 \cdot f(17^2) + 17 \cdot f(17) + 17^2 \cdot f(1) = 1$.Substituting $f(17^2) = 1 - 17^2$ , $f(17) = -16$ , and $f(1) = 1$ : $f(17^2) + 17 \cdot (-16) + 17^2 \cdot 1 = 1, f(17^2) - 272 + 289 = 1, f(17^2) + 17 = 1, f(17^2) = 1 - 17 = -16$.So:$f(2023) = f(7) \cdot f(17^2) = -6 \cdot -16 = 96$.Thus, the value of $f(2023)$ is: $\boxed{96}$. (Correct Answer)

Figure 22: Re$^2$ Examples 4

