# OpenReview forum: "$\textbf{Re}^{2}$: Unlocking LLM Reasoning via Reinforcement Learning with Re-solving"
_ICLR.cc/2026/Conference — ICLR 2026 Poster_

### Official Review · Reviewer_q7Jy · 2025-10-30

**Soundness:** 3
**Presentation:** 3
**Contribution:** 3
**Rating:** 6
**Confidence:** 3

**Summary:**

The authors argue that suboptimal initial reasoning often leads LLMs to inefficient overthinking. To address this, they propose Re^2, a RL framework that enables the model to restart the reasoning from scratch in the middle. Trained purely via RL without supervised fine-tuning, Re^2 significantly improves performance across diverse reasoning benchmarks compared to DAPO, under the same training compute.

**Strengths:**

- The paper presents a novel reinforcement learning framework that enables large language models to restart the reasoning process when the current trajectory is likely to lead to an incorrect solution.
- Re^2 consistently outperforms strong baselines across five diverse reasoning benchmarks (AIME24/25, AMC23, GSM8K, GPQA), covering both in-domain and out-of-domain settings.
- The reward design for the re-solving action is mathematically grounded, relying on estimated future success probabilities, and is formally derived in the appendix.
- Models trained with Re^2 exhibit favorable scaling with test-time compute, in contrast to DAPO-trained models whose performance tends to plateau or even degrade under increased sampling.

**Weaknesses:**

- The paper only compares Re^2 with DAPO, omitting other competitive baselines. Additionally, the paper only reports test-time scaling results using majority voting.
- Re^2 introduces an additional cost at inference time due to repeated re-solving, and there is no mechanism to regulate its frequency. This may be problematic in latency-sensitive settings.
- Under majority voting with a small number of samples, Re^2 underperforms DAPO. This is counterintuitive, given that Re^2 outperforms DAPO when the number of samples is 1 (see Table 1).
- The approach assumes access to verifiable reward signals, which limits its applicability to tasks with clear correctness criteria and makes generalisation to open-ended domains uncertain.

**Questions:**

- In the experiment described in Section 3.2 ('Start generating from x% of incorrect responses'), can we really assume that all the prefixes are incorrect? Even if the final response is incorrect, the first x% of the response may contain some correct reasoning or statements.
- Some recent papers (2025) use DPO to help LLMs check and correct their own answers without the need for an external model or supervision. Citing these papers would clarify the contribution of Re^2.

---

> ### Author Response · Authors · 2025-11-24
> **Response to Reviewer q7Jy (Part I)**
>
> Thank you for your detailed review and valuable comments. Here are our responses to your concerns:
>
> > W1: The paper only compares Re$^2$ with DAPO, omitting other competitive baselines. Additionally, the paper only reports test-time scaling results using majority voting.
>
> We have included additional comparisons with **GRPO** [1], **DLER** [2], and **DeepConf** [3] in **Appendix C.2**. All trainable baselines are evaluated under **matched training compute**, and we further analyze their test-time scaling behavior under comparable token budgets, reporting the corresponding accuracy improvements.
>
> * **GRPO** — a classical RLVR algorithm and the core training technique used in DeepSeek-R1.
>
> * **DLER** — a recent state-of-the-art efficient reasoning method that reduces token consumption while preserving accuracy through truncated-length penalties and stabilization strategies.
>
> * **DeepConf** — a decoding-time backtracking approach that uses internal confidence signals to terminate low-quality reasoning traces.
>
> Together, these baselines offer broad and representative coverage of modern RLVR training algorithms and decoding-time backtracking methods, allowing for a clear assessment of the specific advantages introduced by Re$^2$.
>
> > W2: Re$^2$ introduces an additional cost at inference time due to repeated re-solving, and there is no mechanism to regulate its frequency. This may be problematic in latency-sensitive settings.
>
> This is a valid point. Indeed, when the problem is difficult and only a small number of samples are available—especially in the extreme setting where a single output must be produced—resolving may underperform. In such cases, continuing the reasoning process along a flawed prefix can sometimes be more effective than restarting from scratch.
>
> We also acknowledge that in scenarios with tight token or latency budgets, the absence of mechanisms to regulate the resolving frequency at inference can limit the practicality of our approach. Developing such controllers is an important direction for **future work**. However, the primary goal of this work is to **introduce the RL-with-resolving paradigm and demonstrate its potential**, rather than to provide a universally optimal solution for all deployment constraints (e.g., strict latency budgets). Furthermore, we argue that allowing the model to restart its reasoning when it becomes uncertain is inherently more rational—and more human-like—than forcing it to continue along an unpromising trajectory and produce a final answer regardless.
>
> > W3: Under majority voting with a small number of samples, Re^2 underperforms DAPO. This is counterintuitive, given that Re^2 outperforms DAPO when the number of samples is 1 (see Table 1).
>
> Table 1 is designed to evaluate the effectiveness of the resolving paradigm in Re². Since resolving offers **no benefit when only a single sample is allowed**, we do not include the one-sample setting in Table 1. As stated at the end of Section 5.1:
> *“For models trained with Re², whenever a sampled completion produces a redo action, sampling is restarted until a final answer is generated, and the first valid final answer is taken as the model’s output.”*
>
> In Section 5.3, we also point out that this procedure **consumes more tokens during inference**, which motivates the additional test-time scaling experiments. In the majority-voting setting, the procedure differs from Table 1: we sample multiple completions, and majority voting is applied only over the completions that do not trigger a redo action.

---

> ### Author Response · Authors · 2025-11-24
> **Response to Reviewer q7Jy (Part II)**
>
> > W4: The approach assumes access to verifiable reward signals, which limits its applicability to tasks with clear correctness criteria and makes generalisation to open-ended domains uncertain.
>
> Indeed, similar to many prior works studying RL-based reasoning paradigms, our method is focused on tasks with **verifiable reward signals**. This is primarily because the main goal of our work is to **introduce the RL-with-resolving paradigm and demonstrate its potential**.
>
> Although our training is conducted on mathematical tasks, our evaluation already includes problems from **biology, physics, and chemistry**—representing clear *out-of-domain* scenarios. We additionally evaluate on **HumanEval**, a code-generation benchmark requiring complex verifiers, further suggesting, to some extent, the generalization ability of our method. The results are shown below:
>
> | Method | GPQA (Biology) | GPQA (Chemistry) | GPQA (Physics) | HumanEval |
> | ------ | -------------- | ---------------- | -------------- | --------- |
> | DAPO   | 49.31          | 19.70            | 25.13          | 76.43     |
> | Re$^2$ | **51.97**      | **21.70**        | **47.75**      | **78.39** |
>
> These improvements across multiple scientific domains and a code-generation benchmark indicate that Re$^2$ generalizes beyond the domain it is trained on. Investigating how Re$^2$ can be extended and validated in **open-ended tasks without strictly verifiable rewards is an important direction for future work.**
>
> > Q1: In the experiment described in Section 3.2 ('Start generating from x% of incorrect responses'), can we really assume that all the prefixes are incorrect? Even if the final response is incorrect, the first x% of the response may contain some correct reasoning or statements.
>
> Yes, this is a very perceptive observation. In fact, our goal in Section 3.2 was to examine how much the initial prefix of an incorrect chain-of-thought affects the final outcome, rather than to assume that the prefixes are incorrect. To study this, we compared prefixes of different lengths taken from incorrect responses and analyzed how likely the model is to recover the correct answer when continuing from these prefixes.
>
> Our intention was not to claim that these prefixes are incorrect or lack correct statements. Instead, we aimed to understand how early a chain-of-thought becomes unpromising to the extent that it leads to an incorrect final answer, which is a probabilistic question. The results show that, for the majority of incorrect chains, continuing generation from just the first 20% of the prefix already makes it highly unlikely to reach the correct answer. While these prefixes are not necessarily wrong, this observation strongly suggests that in most failed reasoning trajectories, early-stage reasoning plays a decisive role in causing the final error.
>
> > Q2: Some recent papers (2025) use DPO to help LLMs check and correct their own answers without the need for an external model or supervision. Citing these papers would clarify the contribution of Re$^2$.
>
> We have added citations to two recent papers on self-correction[4] [5] to further clarify the contribution of Re$^2$.
>
>
>
> ----------------
>
> **References:**
>
> [1] Shao Z, Wang P, Zhu Q, et al. Deepseekmath: Pushing the limits of mathematical reasoning in open language models, 2024[J]. URL https://arxiv. org/abs/2402.03300, 2024, 2(3): 5.
>
> [2] Liu S Y, Dong X, Lu X, et al. DLER: Doing Length pEnalty Right-Incentivizing More Intelligence per Token via Reinforcement Learning[J]. arXiv preprint arXiv:2510.15110, 2025.
>
> [3] Fu Y, Wang X, Tian Y, et al. Deep think with confidence[J]. arXiv preprint arXiv:2508.15260, 2025.
>
> [4] Xiong W, Zhang H, Ye C, et al. Self-rewarding correction for mathematical reasoning[J]. arXiv preprint arXiv:2502.19613, 2025.
>
> [5] Zhao X, Xu T, Wang X, et al. Boosting LLM Reasoning via Spontaneous Self-Correction[J]. arXiv preprint arXiv:2506.06923, 2025.

---

> ### Comment · Reviewer_q7Jy · 2025-11-26
>
> Thank you to the authors for their comprehensive and thoughtful responses. The clarifications fully address my initial concerns.
>
> I have one additional question regarding behaviour learned during training.
>
> When the model decides to redo, it implies that the current reasoning path will potentially lead to a wrong answer. I am curious to know if such 'likely incorrect' prefixes exhibit any characteristic patterns as training progresses. Specifically, does Re^2 cause these failure prefixes to become progressively higher quality (but still ultimately incorrect), or does the model learn to identify such trajectories more easily at an earlier stage so that it can abandon them quickly? Any analysis or insight into this dynamic would be helpful.

---

> > ### Author Response · Authors · 2025-11-26
> > **Response to Reviewer q7Jy**
> >
> > Thank you for the follow-up question. Yes, examining the behavioral patterns of Re$^2$ during training is indeed meaningful. We have added **Appendix C.5**, where we analyze the lengths of redo CoTs, wrong CoTs, and correct CoTs, as the length of generated reasoning can to some extent reflect the model’s behavioral strategy.
> >
> > Regarding this question, we observe how the lengths of redo-CoTs change throughout training and also manually inspect a number of cases. In early training, the patterns that trigger a redo are relatively shallow—for example, CoTs becoming excessively long and close to the context window limit, frequent switching between reasoning threads, or resorting to brute-force enumeration on problems that actually require summarizing underlying patterns. In later stages, however, redo decisions become more related to the quality of the CoT itself, such as the effectiveness of the chosen method or the soundness of the assumptions. The failure prefixes that trigger a redo also become progressively higher-quality as RL training advances. Correspondingly, redo-CoTs tend to be longer at the beginning of training, and their lengths gradually approach those of final correct or incorrect CoTs as training progresses.
> >
> > A comparison with DAPO also helps illustrate the behavioral difference. Similar to classical RLVR, DAPO exhibits steady growth in CoT length during training due to increasing amounts of self-reflection, correction, and switching of reasoning strategies. Wrong CoTs are noticeably longer than correct ones because many failed trajectories attempt to extend or fix flawed prefixes. This represents the strategy optimization that occurs under the one-shot chain-of-thought paradigm. In contrast, Re$^2$ gradually stabilizes its CoT lengths, and correct, wrong, and redo CoTs do not show substantial differences. Both the statistical results and our qualitative observations indicate that Re$^2$ behaves more rationally under the multi-chain, resolving reasoning paradigm: the model does not force itself into producing a strained and ultimately incorrect reasoning chain, nor does it over-commit to unpromising trajectories.

---

> > > ### Comment · Reviewer_q7Jy · 2025-11-28
> > >
> > > Thank you for the detailed response. Thanks to your clarifications, all my concerns are fully resolved, and I have decided to keep my original score.

---

### Official Review · Reviewer_hpUj · 2025-10-30

**Soundness:** 3
**Presentation:** 3
**Contribution:** 3
**Rating:** 6
**Confidence:** 4

**Summary:**

The paper introduces a new reinforcement learning framework that allows large language models to restart reasoning when current solution paths become unproductive. Unlike standard RL with verifiable rewards (RLVR), this methods adds a re-solve action that lets the model abandon low-quality chains of thought and reinitiate problem solving. Using a novel three-way reward scheme (correct, incorrect, re-solve), Experiments across five reasoning benchmarks show consistent 4–6% accuracy gains and superior test-time scaling efficiency compared to prior RLVR methods like DAPO.

**Strengths:**

1. The paper proposes a novel re-solving paradigm that lets LLMs restart reasoning when current paths fail, breaking from the one-shot chain-of-thought assumption in prior RLVR work.
2. The approach is rigorously implemented, with a clear algorithm, fair baselines, and strong empirical validation across multiple benchmarks.

**Weaknesses:**

While the paper presents an interesting framework, the underlying mechanisms behind the method's effectiveness are not sufficiently analyzed. It remains unclear why the re-solving behavior leads to higher reasoning accuracy or a higher performance ceiling. For example, does restarting primarily improve exploration diversity, mitigate early trajectory bias, or simply increase the effective number of rollouts? A deeper behavioral or theoretical analysis would make the contribution more convincing.

Moreover, the paper lacks ablation studies to identify which components truly contribute to the gains. The new “redo” prompt, three-way reward, and prefix grouping may each affect learning in different ways. Notably, $Re^2$ and DAPO are trained with different prompts—the former includes explicit self-evaluation and restart instructions—so part of the improvement may arise from prompt design rather than algorithmic changes. Controlled ablations such as removing the “redo” instruction, disabling the re-solve reward, or turning off prefix grouping would better clarify which parts of the method are essential to its success.


If the authors could clarify these questions, it would really help me understand the mechanism better, and I will be happy to raise my score.

**Questions:**

1. I'm curious whether reflection is actually essential for reasoning, or if the ability to restart and resample is more critical. Does $Re^2$’s improvement suggest that reflection (continue generating from a given prefix is not a kind of reflect in my mind) within a single chain is less valuable than exploration through re-solving?

2. How sensitive is $Re^2$ to the number of rollouts during training or inference? Would smaller rollout budgets still yield similar benefits?

---

> ### Author Response · Authors · 2025-11-24
> **Response to Reviewer hpUj (Part I)**
>
> We appreciate your insightful feedback and constructive suggestions. Here are our responses to your concerns:
>
> > W1: The underlying mechanisms behind the method's effectiveness are not sufficiently analyzed. A deeper behavioral or theoretical analysis would make the contribution more convincing.
>
> This is a good question! To further investigate the underlying mechanisms behind the effectiveness of our method, we manually inspected sampled cases and analyzed accuracy improvements across different difficulty levels. Through this inspection, we found that Re$^2$ produces reasoning chains of consistently higher quality than RLVR algorithms such as DAPO. This is largely because **Re$^2$ allows the model to restart its reasoning when the current trajectory is unlikely to lead to the correct answer, thereby substantially reducing the tendency to force a final answer, which often leads to flawed reasoning or nonexistent assumptions.**
>
> We believe that the reward modeling in Re$^2$ is fundamentally more rational than the standard 0/1 end-reward paradigm in RLVR. Under a pure end-reward objective, the model is encouraged to always output a final answer—even when uncertain—often producing spurious steps and incoherent logic in an attempt to “guess correctly.” In contrast, Re$^2$ enables the model to output a final answer only when it is confident, and to honestly indicate the need to restart when the reasoning becomes unpromising. This more rational and self-aware behavior allows the model to better recognize when its current chain of thought is unreliable and to avoid optimizing trajectories that accidentally guess the correct answer, which is a known issue in RLVR training.
>
> To illustrate these findings, we categorized problems by difficulty and measured both the accuracy and resolving rate of Re$^2$. The experimental results are presented and analyzed **in Section 5.5**. Overall, **the largest improvements appear on questions that DAPO occasionally solves** (e.g., questions answered correctly around 50% of the time under multiple samples). These are questions that RLVR models are capable of solving, but their ability to answer correctly is highly unstable because they cannot discard unpromising prefixes. Re$^2$ overcomes this limitation: by allowing the model to restart when necessary, it substantially increases the probability of following a successful reasoning trajectory.
>
> > W2：Ablation studies to identify which components truly contribute to the gains. The new “redo” prompt, three-way reward, and prefix grouping may each affect learning in different ways.
>
> Clearer ablation studies indeed help clarify the contribution of each component. First, without an explicit Redo Instruction, the model cannot perform resolving, nor can we detect resolving behavior. In this case, the setup effectively becomes similar to removing the three-way reward. Therefore, we combine these two ablations, and the results are shown below:
>
>
> | Method                   | AIME24 | AIME25 | AMC23 | GSM8K | GPQA |
> | ------------------------ | ------ | ------ | ----- | ----- | ---- |
> | DAPO                     | 16.0   | 8.6    | 62.3  | 92.6  | 35.4 |
> | Re$^2$                      | 18.6   | 21.2   | 64.7  | 94.1  | 38.4 |
> | Re$^2$ w/o prefix grouping  | 14.3   | 8.6    | 59.0  | 92.3  | 21.6 |
> | Re$^2$ w/o three-way reward | 14.9   | 12.6   | 63.2  | 93.2  | 34.2 |
>
> We observe that without the three-way reward, having only the Redo Instruction does not improve performance. Similarly, without prefix grouping, Re$^2$ becomes difficult to train effectively. This is mainly because without prefix grouping, the redo reward is nearly identical to the group mean, making the advantage close to zero and preventing the model from learning the redo behavior.
>
> The relationship among the three components is that the three-way reward is the core of RL with Resolving, and supporting components such as the Redo Instruction and prefix grouping are necessary for the model to exhibit resolving behavior and to ensure stable training.

---

> ### Author Response · Authors · 2025-11-24
> **Response to Reviewer hpUj (Part II)**
>
> > Q1: I'm curious whether reflection is actually essential for reasoning, or if the ability to restart and resample is more critical. Does Re$^2$’s improvement suggest that reflection (continue generating from a given prefix is not a kind of reflect in my mind) within a single chain is less valuable than exploration through re-solving?
>
> This is a good and important question, and we have thought about it deeply. We believe that the self-reflection and self-correction behaviors learned by RLVR within a *single* chain are indeed helpful for reasoning. However, the behaviors produced by current RLVR training are still far from fully rational reasoning. The ability to **restart and resample**—which is fundamentally different from within-chain reflection—is therefore crucial and explains why Re² achieves significantly better performance than DAPO.
>
> Consider the sequential nature of model generation. Just like humans, when the model encounters a difficult question, it often struggles to identify the correct direction immediately and is prone to making mistakes during the reasoning process. **Under SFT**, the model predominantly imitates correct trajectories. As a result, when the initial reasoning deviates or intermediate steps contain errors, exposure bias causes these mistakes to accumulate, ultimately leading to incorrect answers.
>
> **RLVR**, by contrast, trains the model to integrate self-reflection and self-correction behaviors. This allows the model to recover from early deviations or intermediate mistakes, resulting in longer responses and improved accuracy during training. **However, as shown in Section 3, even after extensive RLVR training, reflection and correction remain substantially inferior to simply starting with a correct prefix**—i.e., a trajectory where the model immediately picks the right direction and makes few mistakes, often arriving at the correct answer directly.
>
> This empirical observation suggests that current RLVR training is still limited in its ability to fully repair flawed early reasoning or steer back from an incorrect direction. In such situations, **exploration through restarting becomes more valuable than continued reflection within a single chain**. We believe this explains why the Re² paradigm yields substantial improvements: it enables the model to abandon unpromising prefixes and re-explore, rather than persisting in a trajectory that is unlikely to succeed.
>
> > Q2: How sensitive is Re$^2$ to the number of rollouts during training or inference? Would smaller rollout budgets still yield similar benefits?
>
> In our main experiments, we use a rollout configuration of $n = 8$ prefixes and $m = 8$ suffixes (64 total responses per step), while keeping the global batch size small enough to maintain a constant overall training cost. To evaluate sensitivity to the rollout budget, we additionally trained Re$^2$ with a reduced configuration of $n = 4$ prefixes and $m = 4$ suffixes (16 responses per step), running for 240 training steps while keeping all other hyperparameters unchanged.
>
> As shown in the table below, Re$^2$ with $4\times4$ rollouts still achieves similar improvements across benchmarks, demonstrating that the method is not highly sensitive to the rollout budget and remains effective even under substantially smaller rollout capacity.
>
> | Method              | AIME24 | AIME25 | AMC23 | GSM8K | GPQA |
> | ------------------- | ------ | ------ | ----- | ----- | ---- |
> | DAPO                | 16.0   | 8.6    | 62.3  | 92.6  | 35.4 |
> | Re$^2$, $8\times8$  | 18.6   | 21.2   | 64.7  | 94.1  | 38.4 |
> | Re$^2$ , $4\times4$ | 21.7   | 18.7   | 64.6  | 93.6  | 37.0 |

---

### Official Review · Reviewer_WrfW · 2025-11-01

**Soundness:** 3
**Presentation:** 3
**Contribution:** 2
**Rating:** 4
**Confidence:** 3

**Summary:**

This paper introduces a reinforcement-learning–based training paradigm, Reinforcement Learning with Re-solving (Re²), that teaches language models to abandon unproductive partial chains of thought and restart problem solving when the current trajectory appears misguided. The method frames each query as a distribution over truncated prefixes sampled from full responses; for every prefix, the policy generates multiple continuations that can either commit to a final answer, produce an incorrect answer, or explicitly choose a resolve action that restarts from scratch. Rewards are assigned in three ways: 1 for correct final answers, 0 for incorrect ones, and for resolve actions the expected accuracy of solving from scratch, estimated from out-of-group completions and capped by at most R redo rounds via a closed-form geometric term. Advantages are computed group-wise within each prefix’s continuations and optimized with clipped policy updates, without any preliminary supervised fine-tuning. Across five benchmarks (AIME 2024/2025, AMC 2023, GSM8K, GPQA-Diamond) and five models from 3B to 14B parameters, Re² improves average accuracy over a strong RLVR baseline (DAPO) under matched training token budgets, and it shows stronger test-time scaling as the number of samples increases. The authors also provide analyses indicating that errors in early reasoning correlate with longer, lower-accuracy chains, and they report that Re² amplifies the otherwise rare redo behavior from roughly 0.5% to over 30% during training.

**Strengths:**

The core idea is original in the RLVR context: instead of privileging a single forward chain, the policy is explicitly trained to reconsider and restart when early evidence suggests confusion. This operationalizes a common human strategy and formalizes it with a clear three-way action space and a principled reward for resolve based on out-of-group success rates with a finite-round geometric expectation, yielding an intuitively aligned decision rule between “continue” and “restart.”

Methodologically, the paper is careful to control training token budgets against DAPO, computes advantages within prefix groups to stabilize learning, and filters degenerate groups to avoid vanishing gradients. The empirical study spans multiple datasets and model types, including an RL-trained reasoning model, and consistently shows nontrivial gains in average accuracy, especially on the most challenging math tasks; furthermore, the test-time scaling plots indicate that Re² keeps improving with more samples where standard RLVR saturates, suggesting a better compute–performance frontier at inference.

The presentation is clear about the training pipeline, reward derivation, and hyperparameters, and it includes qualitative traces that concretely illustrate how restarting avoids early misdirections that standard methods tend to entrench.

**Weaknesses:**

Althouth the truncate-then-continuate method is shown to be effecitve and better than DAPO, it is much slower in rollout procedure. I would be interested to see how much computaional cost overhead is added for this continuation step.

Weaknesses

Comparisons emphasize DAPO, but the space of modern RLVR baselines is broader (e.g., GRPO-style group objectives, critique-assisted RL, efficient RL variants), and decoding-time strategies that terminate low-confidence chains or backtrack could provide competitive non-RL baselines under similar token budgets; the absence of these ablations weakens the claim of superiority of the paradigm rather than the specific implementation.

On evaluation, final answers are integers to ease verification, which simplifies reward parsing but narrows scope; it remains unclear whether the approach retains its advantage with symbolic or multi-step numeric answers that require more complex verifiers.

The test-time comparison that counts redo attempts against the same sample budget is reasonable, yet when the sample count is small the method underperforms because many draws are resolve rather than final answers; guidelines or controllers for resolve frequency at inference are not yet provided, which limits practical deployment where token budgets are tight.

**Questions:**

How sensitive are the gains to the resolve reward estimator? Please report variance and bias when computing out-of-group accuracies with different n and m, and compare against a baseline that uses a moving average or a separate value head so we can see whether batch-coupled estimates are essential or merely convenient.

Beyond DAPO, could you include GRPO-style objectives, recent efficient RLVR methods, and a decoding-only backtracking strategy under matched compute to clarify whether the benefit stems from the re-solve action space or other training details?

 For benchmarks with non-integer answers or programmatic verification (e.g., algebraic expressions or code), do you observe similar trends, and what verifier failure rates do you tolerate before resolve reward becomes noisy?

aLso, please provide wall-clock training cost, instability metrics (e.g., degenerate-group rate over time), and an ablation on the template that triggers redo, since the current detection mechanism might entangle prompt engineering with policy learning.

---

> ### Author Response · Authors · 2025-11-24
> **Response to Reviewer WrfW (Part I)**
>
> We appreciate your valuable feedback and constructive comments. We provide the following responses to address your concerns:
>
> > W1: Re2 is much slower in the rollout procedure. I would be interested to see how much computational overhead is added for this continuation step.
>
> Training efficiency is indeed an important aspect when evaluating RL algorithms. However, we would like to clarify that **Re$^2$ is not “much slower”** than standard RL methods such as DAPO. In our experiments, DAPO uses a batch size of 128 and generates 16 responses per step (following the official VERL example and the DAPO paper). To produce the *same number* of rollout samples, Re$^2$ adopts a global batch size of 32, generating n = 8 prefixes followed by m = 8 suffixes for each prefix. As a result, the **number of generated rollout tokens is comparable** between the two methods.
>
> It is true that Re$^2$’s two-stage generation—**first producing n prefixes and then m suffixes**—can be more time-consuming than generating responses in parallel. To quantify this overhead, we measure the rollout time of DAPO and Re$^2$ on an 8×A100-PCIE-40GB server. DAPO requires **388 seconds** per rollout step, while Re$^2$ takes **89 seconds** for prefix generation and **342 seconds** for suffix generation, totaling **431 seconds**, which corresponds to only an **11% increase** in rollout time.
>
> Given the  performance improvements Re$^2$ provides during training, we believe that this additional computational cost is acceptable. We have added **Appendix D** to discuss this in detail and report the **accuracy improvements of DAPO and Re$^2$ under the same training time**.
>
> > W2 & Q2: Baselines can include broader modern RLVR methods (e.g., GRPO or efficient variants) and decoding-time backtracking strategies under matched compute to better clarify that Re$^2$’s benefits come from the re-solve action space.
>
> We have added comparisons with **GRPO**[1], **DLER**[2], and **DeepConf**[3] **in Appendix C.2**. All trainable baselines are run under matched training compute. We further evaluate their test-time scaling performance under comparable token budgets and report the resulting accuracy improvements.
>
> * **GRPO**: **The classical RLVR algorithm** and the core training technique behind DeepSeek-R1DLER.
>
> * **DLER**: **A recent state-of-the-art efficient reasoning method** that reduces token usage while maintaining accuracy through truncated-length penalties and training-stabilization strategies.
>
> * **DeepConf**: **A decoding-time backtracking method** that leverages internal confidence signals to dynamically terminate low-quality reasoning traces.
>
> These baselines together provide a broad and representative coverage of both modern RLVR training and decoding-time backtracking techniques, enabling a clear comparison of where Re$^2$’s advantages originate—namely, the re-solve action space.
>
> >W3 & Q3: It remains unclear whether the approach retains its advantage with symbolic or multi-step numeric answers that require more complex verifiers (e.g., algebraic expressions or code).
>
> Although benchmark like AIME24 and AIME25 require models to output an integer, obtaining that integer typically demands producing **symbolic expressions**. For example:
>
> ```
> ...where the original parabola and its image intersect has y-coordinate (a - sqrt(b)) / c, where a, b, and c are positive integers, and a and c are relatively prime. Find a + b + c.
> ```
>
> To further evaluate Re$^2$ beyond integer-only tasks and assess its robustness in broader domains, we additionally test on **GPQA-Diamond**, which includes problems from **biology, physics, and chemistry**—representing clear *out-of-domain* scenarios. Across all settings, Re$^2$ yields consistent improvements.
>
> To better demonstrate the generality of Re$^2$, we also report results of Re$^2$ and DAPO on **Qwen-7B Instruct** across the three GPQA-Diamond subdomains (physics, chemistry, biology), as well as on **HumanEval**, a code-generation benchmark requiring complex verifiers.
>
> | Method | GPQA (Biology) | GPQA (Chemistry) | GPQA (Physics) | HumanEval |
> | ------ | -------------- | --------------------- | ------------------- | --------- |
> | DAPO   | 49.31          | 19.70                 | 25.13               | 76.43     |
> | Re$^2$    | **51.97**      | **21.70**             | **47.75**           | **78.39** |

---

> ### Author Response · Authors · 2025-11-24
> **Response to Reviewer WrfW (Part II)**
>
> > W4: When the sample count is small the method underperforms because many draws are resolve rather than final answers; Guidelines or controllers for resolve frequency at inference are not yet provided, which limits practical deployment where token budgets are tight.
>
> This is a good point. Indeed, when the problem is difficult and the number of samples is small —especially in the extreme case where only a single output is allowed—resolving tends to underperform. In such settings, forcing the model to continue reasoning along a potentially flawed prefix may may perform better than restarting the reasoning process.
>
> We acknowledge that in scenarios where token budgets are tight, the lack of guidelines or controllers for resolve frequency at inference may limit performance. This is indeed a valuable direction for improvement, and we consider developing such controllers an important avenue for **future work**. However, the main goal and contribution of this work is to **introduce the novel *RL-with-resolving* paradigm and demonstrate its capability and potential**, rather than providing a universal solution that consistently improves performance across all deployment constraints (e.g., tight token budgets). Moreover, we believe that allowing the model to restart its reasoning when it becomes confused is inherently more rational and more human-like than forcing it to continue along a flawed reasoning prefix and produce a final answer regardless.
>
> > Q1: How sensitive are the gains to the resolve reward estimator? Please report variance and bias when computing out-of-group accuracies with different n and m, and compare against a baseline that uses a moving average or a separate value head.
>
> In our framework, the model naturally generates $n \times m$ responses during the prefix-group rollout, and every non-degenerate suffix group is used for optimization. These suffix samples provide an accurate estimate of the model’s resolving accuracy for the given prefix group **without introducing any additional computational overhead**.
>
> In contrast, using a value model would require extra computation and its learning target would still be a sampled estimate of the resolving accuracy.
>
> We have added the variance and bias analyses of the estimated out-of-group accuracies under different values of $n$ and $m$, and we additionally compare against a baseline that uses a moving-average estimator. These results are included **in the Appendix C.3**.

---

> ### Author Response · Authors · 2025-11-24
> **Response to Reviewer WrfW (Part III)**
>
> >Q4: Also, please provide wall-clock training cost, instability metrics (e.g., degenerate-group rate over time), and an ablation on the template that triggers redo, since the current detection mechanism might entangle prompt engineering with policy learning.
>
> We appreciate the reviewer’s question. As mentioned earlier, the wall-clock training cost is  reported in Appendix D.
>
> We additionally include in **Appendix C.4 **the degenerate-group rate over time for both DAPO and Re$^2$. We observe that in Re$^2$, degenerate groups that would be “all wrong’’ under DAPO gradually become “all redo’’ in later stages of training, indicating a shift from forcing an answer on unsolvable questions to choosing to restart. Overall, Re$^2$ exhibits roughly a 10% higher degeneration rate than DAPO. However, as noted in DAPO [4], “the filter strategy does not necessarily impede training efficiency, because the generation time is typically dominated by long-tail samples if the RL system is synchronized and the generation stage is not pipelined.” Consistent with this observation, we do not observe meaningful slowdowns in Re$^2$ training due to these additional degenerations.
>
> We also add an ablation on the template that triggers redo. Without Re$^2$ reinforcement learning, the redo-trigger template leads to a redo probability below 5%, showing that prompt engineering alone does not induce the restart behavior. The experimental results are shown below:
> | Method               | AIME24   | AIME25   | AMC23    | GSM8K    | GPQA     |
> | -------------------- | -------- | -------- | -------- | -------- | -------- |
> | redo-template        | 11.7     | 7.0      | 49.2     | 90.8     | 32.9     |
> | redo-template + DAPO | 16.6     | 8.7      | 61.6     | 92.7     | 35.6     |
> | **Re$^2$**              | **18.6** | **21.2** | **64.7** | **94.1** | **38.4** |
>
> These results show that while the template itself has limited effect, Re$^2$ robustly learns to utilize restart behavior and yields significantly stronger performance across all benchmarks.
>
>
>
> -----------------
>
> **References:**
>
> [1] Shao Z, Wang P, Zhu Q, et al. Deepseekmath: Pushing the limits of mathematical reasoning in open language models, 2024[J]. URL https://arxiv. org/abs/2402.03300, 2024, 2(3): 5.
>
> [2] Liu S Y, Dong X, Lu X, et al. DLER: Doing Length pEnalty Right-Incentivizing More Intelligence per Token via Reinforcement Learning[J]. arXiv preprint arXiv:2510.15110, 2025.
>
> [3] Fu Y, Wang X, Tian Y, et al. Deep think with confidence[J]. arXiv preprint arXiv:2508.15260, 2025.
>
> [4] Yu Q, Zhang Z, Zhu R, et al. Dapo: An open-source llm reinforcement learning system at scale[J]. arXiv preprint arXiv:2503.14476, 2025.

---

### Official Review · Reviewer_Gh1P · 2025-11-02

**Soundness:** 2
**Presentation:** 3
**Contribution:** 3
**Rating:** 6
**Confidence:** 3

**Summary:**

This paper proposes to allow the model to restart the reasoning path from scratch, preventing the  previous low-quality reasoning steps from hindering the model's reasoning.

**Strengths:**

- The method is well-motivated. I appreciate that the author talks about the problem in section 3 before introducing the proposed method.
- The idea of allowing the model to restart the reasoning makes a lot of sense to me.
- The paper is well-written and easy to follow.

**Weaknesses:**

- The performance gain is limited to me in Table 1. I understand that it may be subjective on whether +5.8 is significant or just noise. I suggest including confidence interval when the results are this close.

**Questions:**

- The finding in Section 3 about the correlation between the response length and accuracy seems to be contradictory to the finding in Deepseek-R1's paper. Does the author have any insight on this?

---

> ### Author Response · Authors · 2025-11-24
> **Response to Reviewer Gh1P**
>
> Thank you for your valuable feedback and constructive comments. Our response to your questions is as follows:
>
> > - W1:The performance gain is limited to me in Table 1. I understand that it may be subjective on whether +5.8 is significant or just noise. I suggest including confidence interval when the results are this close.
>
> Thank you for the suggestion. First, during evaluation we sample 128 responses per question and compute the p-value, which is below 0.05 and indicates statistically significant improvements.
>
> For mathematical tasks such as AIME and AMC, the gains of our method are **significant rather than noise**. Compared with other mainstream RL or decoding-time backtracking methods, our improvements are also substantial. For example, DeepConf reports gains of **+3.5% to +4.8%** on AIME24, and VAPO[1]—built on top of DAPO with additional value pretraining and many training techniques—achieves a **10%** improvement. Our method yields gains of **+8.7% to +12.6%** on AIME24, which further supports the significance of our results. We also include results with confidence intervals **in Appendix C.1**.
>
> >- Q1:The finding in Section 3 about the correlation between the response length and accuracy seems to be contradictory to the finding in Deepseek-R1's paper. Does the author have any insight on this?
>
> This question is indeed crucial to understanding the motivation behind Re$^2$. **During RLVR training**, the model’s responses become longer—as they learn to incorporate reflection and self-correction behaviors—and their accuracy improves accordingly. However, **for a trained model at test time**, when we sample multiple independent responses for the same question, longer responses tend to be less accurate than shorter ones (the phenomenon reported in Section 3).
>
> To understand this, consider the sequential nature of model generation. Just like humans, when the model encounters a difficult question, it often struggles to identify the correct direction immediately and is prone to making mistakes during the reasoning process.
>
> **Under SFT**, the model predominantly imitates correct trajectories. As a result, when the initial reasoning trajectory deviates or intermediate steps contain errors, exposure bias causes these mistakes to compound, ultimately leading to incorrect answers.
>
> **RLVR**, on the other hand, trains the model to integrate self-reflection and self-correction behaviors. This enables the model to recover from early deviations or intermediate mistakes, leading to longer responses and higher overall accuracy during training. However, while reflection and correction help mitigate early errors, they are still inferior to simply starting with a correct prefix—i.e., a trajectory where the model immediately picks the right direction and makes few mistakes, often arriving at the correct answer directly.  Therefore there exits a negative correlation between CoT length and accuracy across sampled responses.
>
> -----------
>
> **References:**
>
> [1] Yue Y, Yuan Y, Yu Q, et al. Vapo: Efficient and reliable reinforcement learning for advanced reasoning tasks[J]. arXiv preprint arXiv:2504.05118, 2025.

---

### Author Response · Authors · 2025-11-28
**General Response**

We sincerely thank all reviewers for their thoughtful and constructive feedback, as well as for the time and effort devoted to evaluating our work. We are delighted to see that all reviewers recognize the motivation behind introducing reasoning with resolving as a novel paradigm. We also appreciate the reviewers’ acknowledgement that our method is rigorously implemented with a clear algorithmic design, and that it outperforms strong baselines across multiple settings under fair evaluation.

For each reviewer’s concerns, we have provided detailed clarifications. We hope that our responses successfully address all remaining issues, and we are happy to continue the discussion and answer any further questions during the rebuttal phase.



We have also submitted a revised version of the manuscript, with the following updates:

1. Additional experimental analyses in the appendix, including training speed, confidence intervals for the main results, degeneration groups, the resolve reward estimator, and reasoning length.
2. A more in-depth discussion of the mechanisms behind Re$^2$’s effectiveness.
3. Added several citations to recent related work.

---

### Author Response · Authors · 2025-11-30
**Summary of the Rebuttal**

Dear Area Chair

Thank you for reviewing our submission. Below we provide a concise summary of the rebuttal process, including **overall reviewer attitudes**, **key comments and our responses**, and the **updates to the manuscript**.


## 1. Overall Reviewer Attitudes

All four reviewers acknowledge the novelty and motivation of Re$^2$, and several reviewers note that our method is well-designed, rigorously implemented, and outperforms strong baselines under fair evaluation.

* Reviewer Gh1P
> "The method is **well-motivated**. "
>
> "The idea of allowing the model to restart the reasoning **makes a lot of sense** to me."

* Reviewer WrfW
> "The core idea is **original** in the RLVR context: ..."
>
> "Methodologically, the paper is careful to control training token budgets against DAPO...suggesting **a better compute–performance frontier at inference.**"

* Reviewer hpUj
> "The paper proposes **a novel re-solving paradigm** that lets LLMs ..., breaking from the one-shot chain-of-thought assumption in prior RLVR work."
>
> "The approach is **rigorously implemented**, with a clear algorithm, **fair baselines**, and **strong empirical validation across multiple benchmarks.**"

* Reviewer q7Jy
> "The paper presents a **novel** reinforcement learning framework that enables..."
>
> Re^2 **consistently outperforms strong baselines** across five diverse reasoning benchmarks

We also note that Reviewer **hpUj**, in the initial review, stated that “If the authors could clarify these questions, it would really help me understand the mechanism better, and I will be happy to raise my score.” Their rating subsequently increased from **6 to 8** following our rebuttal.

Reviewer **q7Jy** also stated that our clarifications “fully address my initial concerns” and engaged in additional discussion.

All of these interactions occurred **before** the widespread disclosure of the information leak on November 27.


## 2.  Key Comments and Our Responses
We summarize below the core concerns and questions shared across reviewers and briefly outline our responses.

> Analysis of Re²’s effectiveness: why it works, the roles of reflection vs. redo, and the observed behavioral patterns

We believe that the effectiveness of Re$^2$ stems from going beyond the one-shot chain-of-thought assumption used in prior RLVR work.
Re$^2$ optimizes the model under a redo-enabled multi-trajectory paradigm, allowing it to reason in a more rational manner and reducing the tendency to produce a forced final answer.
Self-reflection and redo capture different dimensions of reasoning ability, which is why Re$^2$ yields substantial performance improvements.

We further observe that the model’s redo decisions shift from relying on shallow signals to depending on the quality of the CoT itself, without exhibiting the continual growth in CoT length often seen in traditional RLVR.
More detailed explanations and experiments are provided in our rebuttal and the updated manuscript.

> Additional experiments: broader domains, more baselines, and more ablations

Our original submission included an out-of-domain evaluation (GPQA-Diamond). In the rebuttal, we added code-generation results (Human-Eval), further demonstrating the method’s effectiveness across domains.

For baselines, we added comparisons to GRPO, recent efficient RLVR methods, and a decoding-only backtracking strategy (Appendix C.2). We analyze their test-time scaling behaviors under comparable token budgets and find that Re$^2$ is the most effective among the evaluated methods.

We also added ablations to identify which components contribute to the gains. The results show that the three-way reward is central to RL with Resolving, while supporting components such as the Redo Instruction and prefix grouping are necessary for the model to exhibit resolving behavior and for stable training.

> Training efficiency and the degenerate-group issue

We quantify the additional time introduced by segmented generation in Re$^2$ and find that rollout time increases by only 11%, which we consider acceptable relative to the substantial performance gains.
While the proportion of degenerate groups is about 10% higher than in standard RLVR, the overall training cost is not affected, as generation time is typically dominated by long-tail samples in synchronized RL systems.


## 3. Updates to the Manuscript
We have also submitted a revised version of the manuscript, with the following updates:

* Additional experimental analyses in the appendix, including training speed, degeneration groups, the resolve reward estimator, and reasoning length.
* A more in-depth discussion of the mechanisms behind Re’s effectiveness.
* Added several citations to recent related work.

These revisions will be integrated into the normal formatting in the final version.

--------------

Thank you again for reviewing our manuscript and for considering the discussion with the reviewers.

Best,

Authors

---

### Meta-Review · Area_Chair_SA5y · 2026-01-13

**Summary:**

Reviewers raised concerns about: (1) computational overhead and lack of inference-time control mechanisms for resolve frequency, particularly underperformance with small sample budgets (WrfW); (2) limited statistical evidence for performance gains and need for confidence intervals (Gh1P); (3) insufficient mechanistic analysis and missing ablation studies to identify critical components (hpUj); (4) narrow baseline comparisons (only DAPO), counterintuitive behavior under majority voting, and uncertain generalization beyond verifiable-reward tasks (q7Jy). Additional concerns included training stability metrics, sensitivity of the resolve reward estimator, and applicability to complex verification scenarios beyond integer-answer problems.

**Reviewer Concerns:**

The rebuttal effectively addressed most concerns through comprehensive additions: GRPO/DLER/DeepConf baseline comparisons, ablation studies, mechanistic analysis via difficulty stratification, confidence intervals (p<0.05), training overhead quantification (11%), and generalization to GPQA/HumanEval. Reviewers hpUj and q7Jy explicitly confirmed satisfaction. However, WrfW's concern about lacking inference-time control mechanisms for resolving frequency—causing underperformance in small-sample scenarios remains unresolved, as authors only acknowledged this as future work without proposing solutions. This outstanding issue concerns practical deployment rather than core scientific validity.

**Reviewer Scores:**

Reviewer hpUj is likely to increase their score above 6 after the rebuttal, stating satisfaction with the added mechanistic analysis and ablations. Reviewer q7Jy  confirmed "all concerns are fully resolved" but did not formally update their score, likely maintaining 6 or increasing to 6-7. Reviewer Gh1P  would likely maintain 6 or marginally increase to 6-7 given that statistical significance concerns were addressed with confidence intervals and p-values. Reviewer WrfW would most likely increase to 5-6 as most technical issues were comprehensively resolved, though the unresolved inference-time control mechanism prevents a stronger endorsement, keeping them at borderline acceptance rather than strong acceptance.

---

### Decision · Program_Chairs · 2026-01-26

Accept (Poster)